# Online Clustering of Dueling Bandits

**Zhiyong Wang** [1]  **Jiahang Sun** [2]  **Mingze Kong** [3]  **Jize Xie** [4]  **Qinghua Hu** [5]  **John C.S. Lui** [1]  **Zhongxiang Dai** [3]

## Abstract

The contextual multi-armed bandit (MAB) is a widely used framework for problems requiring sequential decision-making under uncertainty, such as recommendation systems. In applications involving a large number of users, the performance of contextual MAB can be significantly improved by facilitating collaboration among multiple users. This has been achieved by the clustering of bandits (CB) methods, which adaptively group the users into different clusters and achieve collaboration by allowing the users in the same cluster to share data. However, classical CB algorithms typically rely on numerical reward feedback, which may not be practical in certain real-world applications. For instance, in recommendation systems, it is more realistic and reliable to solicit *preference feedback* between pairs of recommended items rather than absolute rewards. To address this limitation, we introduce the first "clustering of dueling bandit algorithms" to enable collaborative decision-making based on preference feedback. We propose two novel algorithms: (1) Clustering of Linear Dueling Bandits (COLDB) which models the user reward functions as linear functions of the context vectors, and (2) Clustering of Neural Dueling Bandits (CONDB) which uses a neural network to model complex, non-linear user reward functions. Both algorithms are supported by rigorous theoretical analyses, demonstrating that user collaboration leads to improved regret bounds. Extensive empirical evaluations on synthetic and real-world datasets further validate the effectiveness of our methods, establishing their potential in real-world applications involving multiple users with preference-based feedback.

[1]The Chinese University of Hong Kong [2]Tongji University [3]The Chinese University of Hong Kong, Shenzhen [4]Hong Kong University of Science and Technology [5]Tianjin University. Correspondence to: Zhongxiang Dai <daizhongxiang@cuhk.edu.cn>.

*Proceedings of the $42^{nd}$ International Conference on Machine Learning*, Vancouver, Canada. PMLR 267, 2025. Copyright 2025 by the author(s).

## 1. Introduction

The contextual multi-armed bandit (MAB) is a widely used method in real-world applications requiring sequential decision-making under uncertainty, such as recommendation systems, computer networks, among others (Li et al., 2010; Wang et al., 2023; Dai et al., 2024b). In a contextual MAB problem, a user faces a set of $K$ arms (i.e., context vectors) in every round, selects one of these $K$ arms, and then observes a corresponding numerical reward (Lattimore & Szepesvári, 2020). In order to select the arms to maximize the cumulative reward (or equivalently minimize the cumulative regret), we often need to consider the trade-off between the *exploration* of the arms whose unknown rewards are associated with large uncertainty and *exploitation* of the available observations collected so far. To carefully handle this trade-off, we often model the reward function using a surrogate model, such as a linear model (Chu et al., 2011) or a neural network (Zhou et al., 2020).

Some important applications of contextual MAB, such as recommendation systems, often involve a large number (e.g., in the scale of millions) of users, which opens up the possibility of further improving the performance of contextual MAB via user collaboration. To this end, the method of *online Clustering of Bandits* (CB) has been proposed, which adaptively partitions the users into a number of clusters and leverages the collaborative effect of the users in the same cluster to achieve improved performance (Gentile et al., 2014; Wang et al., 2024a; Li et al., 2019).

Classical CB algorithms usually require an absolute real-valued numerical reward as feedback for each arm (Wang et al., 2024a). However, in some crucial applications of contextual MAB, it is often more realistic and reliable to request the users for *preference feedback*. For example, in recommendation systems, it is often preferable to recommend a pair of items to a user and then ask the user for relative feedback (i.e., which item is preferred) (Yue et al., 2012). As another example, contextual MAB has been successfully adopted to optimize the input prompt for large language models (LLMs), which is often referred to as *prompt optimization* (Lin et al., 2024a;b). In this application, instead of requesting an LLM user for a numerical score as feedback, it is more practical to show the user a pair of LLM responses generated by two candidate prompts and ask the user which

response is preferred (Lin et al., 2024a; Verma et al., 2024).

A classical and principled approach to account for preference feedback in contextual MAB is the framework of contextual *dueling bandit* (Saha, 2021; Bengs et al., 2022; Saha & Krishnamurthy, 2022; Li et al., 2024). In every round of contextual dueling bandits, a pair of arms are selected, after which a binary observation is collected reflecting which arm is preferred. However, classical dueling bandit algorithms are not able to leverage the collaboration of multiple users, which leaves significant untapped potential to further improve the performance in these applications involving preference feedback. In this work, we bring together the merits of both approaches, and hence introduce the first *clustering of dueling bandit* algorithms, enabling multi-user collaboration in scenarios involving preference feedback.

We firstly proposed our *Clustering Of Linear Dueling Bandits* (COLDB) algorithm (Sec. 3.1), which assumes that the latent reward function of each user is a linear function of the context vectors (i.e., the arm features). In addition, to handle challenging real-world scenarios with complicated non-linear reward functions, we extend our COLDB algorithm to use a *neural network to model the reward function*, hence introducing our *Clustering Of Neural Dueling Bandits* (CONDB) algorithm (Sec. 3.2). Both algorithms adopt a graph to represent the estimated clustering structure of all users, and adaptively update the graph to iteratively refine the estimate. After receiving a user in every round, our both algorithms firstly assign the user to its estimated cluster, and then leverage the data from all users in the estimated cluster to learn a linear model (COLDB) or a neural network (CONDB), which is then used to select a pair of arms for the user to query for preference feedback. After that, we update the reward function estimate for the user based on the newly observed feedback, and then update the graph to remove its connection with users who are estimated to belong to a different cluster.

We conduct rigorous theoretical analysis for both our COLDB and CONDB algorithms, and our theoretical results demonstrate that the regret upper bounds of both algorithms are sub-linear and that a larger degree of user collaboration (i.e., when a larger number of users belong to the same cluster on average) leads to theoretically guaranteed improvement (Sec. 4). In addition, we also perform both synthetic and real-world experiments to demonstrate the practical advantage of our algorithms and the benefit of user collaboration in contextual MAB problems with preference feedback (Sec. 5).

## 2. Problem Setting

This section formulates the problem of *clustering of dueling bandits*. In the following, we use boldface lowercase letters for vectors and boldface uppercase letters for matrices. The number of elements in a set $\mathcal{A}$ is denoted as $|\mathcal{A}|$, while $[m]$ refers to the index set $\{1, 2, \ldots, m\}$, and $\|\boldsymbol{x}\|_{\boldsymbol{M}} = \sqrt{\boldsymbol{x}^\top \boldsymbol{M} \boldsymbol{x}}$ represents the matrix norm of vector $\boldsymbol{x}$ with respect to the positive semi-definite (PSD) matrix $\boldsymbol{M}$.

**Clustering Structure.** Consider a scenario with $u$ users, indexed by $\mathcal{U} = \{1, 2, \ldots, u\}$, where each user $i \in \mathcal{U}$ is associated with a unknown reward function $f_i : \mathbb{R}^{d'} \to \mathbb{R}$ which maps an arm $\boldsymbol{x} \in \mathcal{X} \subset \mathbb{R}^{d'}$ to its corresponding reward value $f_i(\boldsymbol{x})$. We assume that there exists an underlying, yet unknown, clustering structure over the users reflecting their behavior similarities. Specifically, the set of users $\mathcal{U}$ is partitioned into $m$ clusters $C_1, C_2, \ldots, C_m$, where $m \ll u$, and the clusters are mutually disjoint: $\cup_{j \in [m]} C_j = \mathcal{U}$ and $C_j \cap C_{j'} = \emptyset$ for $j \neq j'$. These clusters are referred to as *ground-truth clusters*, and the set of clusters is denoted by $\mathcal{C} = \{C_1, C_2, \ldots, C_m\}$. Let $f^j$ denote the common reward function of all users in cluster $j$ and let $j(i) \in [m]$ be the index of the cluster to which user $i$ belongs. If two users $i$ and $l$ belong to the same cluster, they have the same reward function. That is, for any $\ell \in \mathcal{U}$, if $\ell \in C_{j(i)}$, then $f_\ell = f_i = f^{j(i)}$. Meanwhile, users from different clusters have distinct reward functions.

**Modeling Preference Feedback.** At each time step $t \in [T]$, a user $i_t \in \mathcal{U}$ is served. The learning agent observes a set of context vectors (i.e., arms) $\mathcal{X}_t \subseteq \mathcal{X} \subset \mathbb{R}^{d'}$, where $|\mathcal{X}_t| = K \leq C$ for all $t$. Each arm $\boldsymbol{x} \in \mathcal{X}_t$ is a feature vector in $\mathbb{R}^{d'}$ with $\|\boldsymbol{x}\|_2 \leq 1$. The agent assigns the cluster $\overline{C}_t$ to user $i_t$ and recommends two arms $\boldsymbol{x}_{t,1}, \boldsymbol{x}_{t,2} \in \mathcal{X}_t$ based on the aggregated historical data from cluster $\overline{C}_t$. After receiving the recommended pair of arms, the user provides a binary preference feedback $y_t \in \{0, 1\}$, in which $y_t = 1$ if $\boldsymbol{x}_{t,1}$ is preferred over $\boldsymbol{x}_{t,2}$ and $y_t = 0$ otherwise. We model the binary preference feedback following the widely used Bradley-Terry-Luce (BTL) model (Hunter, 2004; Luce, 2005). Specifically, the BTL model assumes that for user $i_t$, the probability that the first arm $\boldsymbol{x}_{t,1}$ is preferred over the second arm $\boldsymbol{x}_{t,2}$ is given by

$$\mathbb{P}_t(\boldsymbol{x}_{t,1} \succ \boldsymbol{x}_{t,2}) = \mu(f_{i_t}(\boldsymbol{x}_{t,1}) - f_{i_t}(\boldsymbol{x}_{t,2})),$$

where $\mu : \mathbb{R} \to [0, 1]$ is the logistic function: $\mu(z) = \frac{1}{1+e^{-z}}$. In other words, the binary feedback $y_t$ is sampled from the Bernoulli distribution with the probability $\mathbb{P}_t(\boldsymbol{x}_{t,1} \succ \boldsymbol{x}_{t,2})$.

We make the following assumption about the preference model:

**Assumption 2.1** (Standard Dueling Bandits Assumptions).
1. $|\mu(f(\boldsymbol{x})) - \mu(g(\boldsymbol{x}))| \leq L_\mu |f(\boldsymbol{x}) - g(\boldsymbol{x})|, \forall x \in \mathcal{X}$, for any functions $f, g : \mathbb{R}^{d'} \to \mathbb{R}$.
2. $\min_{\boldsymbol{x} \in \mathcal{X}} \nabla \mu(f(\boldsymbol{x})) \geq \kappa_\mu > 0$.

Assumption 2.1 is the standard assumption in the analysis of linear bandits and dueling bandits (Li et al., 2017; Bengs

et al., 2022), and when $\mu$ is the logistic function, $L_\mu = 1/4$. The regret incurred by the learning agent is defined as:

$$R_T = \sum_{t=1}^{T} r_t = \sum_{t=1}^{T} \left(2f_{i_t}(\boldsymbol{x}_t^*) - f_{i_t}(\boldsymbol{x}_{t,1}) - f_{i_t}(\boldsymbol{x}_{t,2})\right),$$

where $\boldsymbol{x}_t^* = \arg\max_{\boldsymbol{x} \in \mathcal{X}_t} f_{i_t}(\boldsymbol{x})$ represents the optimal arm at round $t$. This is a commonly adopted notion of regret in the analysis of dueling bandits (Bengs et al., 2022; Saha & Krishnamurthy, 2022).

## 2.1. Clustering of Linear Dueling Bandits

For the linear setting, we assume that each reward function $f_i$ is linear in a fixed feature space $\phi(\cdot)$, such that $f_i(\boldsymbol{x}) = \boldsymbol{\theta}_i^\top \phi(\boldsymbol{x}), \forall \boldsymbol{x} \in \mathcal{X}$. The feature mapping $\phi : \mathbb{R}^{d'} \to \mathbb{R}^d$ is a fixed mapping with $\|\phi(\boldsymbol{x})\|_2 \leq 1$ for all $\boldsymbol{x} \in \mathcal{X}$. In the special case of classical linear dueling bandits, we have that $\phi(\boldsymbol{x}) = \boldsymbol{x}$, i.e., $\phi(\cdot)$ is the identity mapping. The use of $\phi(\boldsymbol{x})$ enables us to potentially model non-linear reward functions given an appropriate feature mapping.

In this case, the reward function of every user $i$ is represented by its corresponding *preference vector* $\boldsymbol{\theta}_i$, and all users in the same cluster share the same preference vector while users from different clusters have distinct preference vectors. Denote $\boldsymbol{\theta}^j$ as the common preference vector of users in cluster $C_j$, and let $j(i) \in [m]$ be the index of the cluster to which user $i$ belongs. Therefore, for any $\ell \in \mathcal{U}$, if $\ell \in C_{j(i)}$, then $\boldsymbol{\theta}_\ell = \boldsymbol{\theta}_i = \boldsymbol{\theta}^{j(i)}$.

The following assumptions are made regarding the clustering structure, users, and items:

**Assumption 2.2** (Cluster Separation). The preference vectors of users from different clusters are at least separated by a constant gap $\gamma > 0$, i.e.,

$$\left\|\boldsymbol{\theta}^j - \boldsymbol{\theta}^{j'}\right\|_2 \geq \gamma \quad \text{for all} \quad j \neq j' \in [m].$$

**Assumption 2.3** (Uniform User Arrival). At each time step $t$, the user $i_t$ is selected uniformly at random from $\mathcal{U}$, with probability $1/u$, independent of previous rounds.

**Assumption 2.4** (Item regularity). At each time step $t$, the feature vector $\phi(\boldsymbol{x})$ of each arm $\boldsymbol{x} \in \mathcal{X}_t$ is drawn independently from a fixed but unknown distribution $\rho$ over $\{\phi(\boldsymbol{x}) \in \mathbb{R}^d : \|\phi(\boldsymbol{x})\|_2 \leq 1\}$, where $\mathbb{E}_{\boldsymbol{x}_1, \boldsymbol{x}_2 \sim \rho}[(\phi(\boldsymbol{x}_1) - \phi(\boldsymbol{x}_2))(\phi(\boldsymbol{x}_1) - \phi(\boldsymbol{x}_2))^\top]$ is full rank with minimal eigenvalue $\lambda_x > 0$. Additionally, at any time $t$, for any fixed unit vector $\boldsymbol{\theta} \in \mathbb{R}^d$, $(\boldsymbol{\theta}^\top(\phi(\boldsymbol{x}_1) - \phi(\boldsymbol{x}_2)))^2$ has sub-Gaussian tail with variance upper bounded by $\sigma^2$.

**Remark 1.** All these assumptions above follow the previous works on clustering of bandits (Gentile et al., 2014; 2017; Li & Zhang, 2018; Ban & He, 2021b; Liu et al., 2022; Wang et al., 2024a;b). For Assumption 2.3, our results can easily generalize to the case where the user arrival follows any distribution with minimum arrival probability $\geq p_{min}$.

## 2.2. Clustering of Neural Dueling Bandits

Here we allow the reward functions $f_i$'s to be non-linear functions. To estimate the unknown reward functions $f_i$'s, we use fully connected neural networks (NNs) with ReLU activations, and denote the depth and width (of every layer) of the NN by $L \geq 2$ and $m_{\text{NN}}$, respectively (Zhou et al., 2020; Zhang et al., 2021). Let $h(\boldsymbol{x}; \boldsymbol{\theta})$ represent the output of an NN with parameters $\boldsymbol{\theta}$ and input vector $\boldsymbol{x}$, which is defined as follows:

$$h(\boldsymbol{x}; \boldsymbol{\theta}) = \mathbf{W}_L \text{ReLU}\left(\mathbf{W}_{L-1}\text{ReLU}\left(\cdots \text{ReLU}\left(\mathbf{W}_1 \boldsymbol{x}\right)\right)\right),$$

in which $\text{ReLU}(\boldsymbol{x}) = \max\{\boldsymbol{x}, 0\}$, $\mathbf{W}_1 \in \mathbb{R}^{m_{\text{NN}} \times d}$, $\mathbf{W}_l \in \mathbb{R}^{m_{\text{NN}} \times m_{\text{NN}}}$ for $2 \leq l < L$, $\mathbf{W}_L \in \mathbb{R}^{1 \times m_{\text{NN}}}$. We denote the parameters of NN by $\boldsymbol{\theta} = (\text{vec}(\mathbf{W}_1); \cdots \text{vec}(\mathbf{W}_L))$, where $\text{vec}(A)$ converts an $M \times N$ matrix $A$ into a $MN$-dimensional vector. We use $p$ to denote the total number of NN parameters: $p = dm_{\text{NN}} + m_{\text{NN}}^2(L-1) + m_{\text{NN}}$, and use $g(\boldsymbol{x}; \boldsymbol{\theta})$ to denote the gradient of $h(\boldsymbol{x}; \boldsymbol{\theta})$ with respect to $\boldsymbol{\theta}$.

The algorithmic design and analysis of neural bandit algorithms make use of the theory of the *neural tangent kernel* (NTK) (Jacot et al., 2018). We let all $u$ users use the same initial NN parameters $\boldsymbol{\theta}_0$, and assume that the value of the *empircal NTK* is bounded: $\frac{1}{m_{\text{NN}}} \langle g(\boldsymbol{x}; \boldsymbol{\theta}_0), g(\boldsymbol{x}; \boldsymbol{\theta}_0) \rangle \leq 1, \forall \boldsymbol{x} \in \mathcal{X}$. This is a commonly adopted assumption in the analysis of neural bandits (Dai et al., 2023; Kassraie & Krause, 2022). Let $T^j$ denote total number of rounds in which the users in cluster $j$ is served. We use $\mathbf{H}_j$ to denote the *NTK matrix* (Zhou et al., 2020) for cluster $j$, which is a $(T_j K) \times (T_j K)$-dimensional matrix. Similarly, we define $\mathbf{h}_j$ as the $(T_j K) \times 1$-dimensional vector containing the reward function values of all $T_j K$ arm feature vectors for cluster $j$. We provide the concrete definitions of $\mathbf{H}_j$ and $\mathbf{h}_j$ in App. C.1. We make the following assumptions which are commonly adopted by previous works on neural bandits (Zhou et al., 2020; Zhang et al., 2021), for which we provide justifications in App. C.1.

**Assumption 2.5.** The reward functions for all users are bounded: $|f_i(x)| \leq 1, \forall x \in \mathcal{X}, \forall i \in \mathcal{U}$. There exists $\lambda_0 > 0$ s.t. $\mathbf{H}_j \succeq \lambda_0 I, \forall j \in \mathcal{C}$. All arm feature vectors satisfy $\|x\|_2 = 1$ and $x_j = x_{j+d/2}, \forall x \in \mathcal{X}_t, \forall t \in [T]$.

Denote by $f^j$ the common reward function of the users in cluster $C_j$, and let $j(i) \in [m]$ be the index of the cluster to which user $i$ belongs. Same as Sec. 2.1, here all users in the same cluster share the same reawrd function. Therefore, for any $\ell \in \mathcal{U}$, if $\ell \in C_{j(i)}$, then $f_\ell(\boldsymbol{x}) = f_i(\boldsymbol{x}) = f^{j(i)}(\boldsymbol{x}), \forall \boldsymbol{x} \in \mathcal{X}$. The following lemma shows that when the NN is wide enough (i.e., $m_{\text{NN}}$ is large), the reward function of every cluster can be modeled by a linear function.

**Lemma 2.6** (Lemma B.3 of (Zhang et al., 2021)). *As long as the width $m_{NN}$ of the NN is large enough: $m_{NN} \geq$*

$poly(T, L, K, 1/\kappa_\mu, L_\mu, 1/\lambda_0, 1/\lambda, \log(1/\delta))$, *then for all clusters* $j \in [m]$, *with probability of at least* $1 - \delta$, *there exits a* $\boldsymbol{\theta}_f^j$ *such that*

$$f^j(\boldsymbol{x}) = \langle g(\boldsymbol{x}; \boldsymbol{\theta}_0), \boldsymbol{\theta}_f^j - \boldsymbol{\theta}_0 \rangle,$$

$$\sqrt{m_{NN}} \left\| \boldsymbol{\theta}_f^j - \boldsymbol{\theta}_0 \right\|_2 \le \sqrt{2\mathbf{h}_j^\top \mathbf{H}_j^{-1} \mathbf{h}_j} \le B,$$

*for all* $\boldsymbol{x} \in \mathcal{X}_t$, $t \in [T]$ *with* $i_t \in C_j$.

We provide the detailed statement of Lemma 2.6 in Lemma C.1 (App. C.2). For a user $i$ belonging to cluster $j(i)$, we let $\boldsymbol{\theta}_{f,i} = \boldsymbol{\theta}_f^{j(i)}$, then we have that $f_i(\boldsymbol{x}) = \langle g(\boldsymbol{x}; \boldsymbol{\theta}_0), \boldsymbol{\theta}_{f,i} - \boldsymbol{\theta}_0 \rangle, \forall \boldsymbol{x} \in \mathcal{X}$. As a result of Lemma 2.6, for any $\ell \in \mathcal{U}$, if $\ell \in C_{j(i)}$, we have that $\boldsymbol{\theta}_{f,\ell} = \boldsymbol{\theta}_{f,i} = \boldsymbol{\theta}^{j(i)}, \forall \boldsymbol{x} \in \mathcal{X}$.

The assumption below formalizes the gap between different clusters in a similar way to Assumption 2.2.

**Assumption 2.7** (Cluster Separation)**.** The reward functions of users from different clusters are separated by a constant gap $\gamma'$:

$$\left\| f^j(\boldsymbol{x}) - f^{j'}(\boldsymbol{x}) \right\|_2 \ge \gamma' > 0, \forall j, j' \in [m], j \ne j' \, \forall \boldsymbol{x} \in \mathcal{X}.$$

In neural bandits, we adopt $(1/\sqrt{m_{NN}})g(\boldsymbol{x}; \boldsymbol{\theta}_0)$ as the feature mapping. Therefore, our item regularity assumption (Assumption 2.4) is also applicable here after plugging in $\phi(\boldsymbol{x}) = (1/\sqrt{m_{NN}})g(\boldsymbol{x}; \boldsymbol{\theta}_0)$.

# 3. Algorithms

### 3.1. Clustering Of Linear Dueling Bandits (COLDB)

Our Clustering Of Linear Dueling Bandits (COLDB) algorithm is described in Algorithm 1. Here we elucidate the underlying principles and operational workflow of COLDB. COLDB maintains a dynamic graph $G_t = (\mathcal{U}, E_t)$ encompassing all users, whose connected components represent the inferred user clusters in round $t$. Throughout the learning process, COLDB adaptively removes edges to accurately cluster the users based on their estimated reward function parameters, thereby leveraging these clusters to enhance online learning efficiency. The operation of COLDB proceeds as follows:

**Cluster Inference** $\overline{C}_t$ **for User** $i_t$ **(Line 2-Line 5).** Initially, COLDB constructs a complete undirected graph $G_0 = (\mathcal{U}, E_0)$ over the user set (Line 2). As learning progresses, edges are selectively removed to ensure that only users with similar preference profiles remain connected. At each round $t$, when a user $i_t$ comes to the system with a feasible arm set $\mathcal{X}_t$ (Line 4), COLDB identifies the connected component $\overline{C}_t$ containing $i_t$ in the maintained graph $G_{t-1}$, which serves as the current estimated cluster for this user (Line 5).

**Estimating Shared Statistics for Cluster** $\overline{C}_t$ **(Line 6-Line 7).** Once the cluster $\overline{C}_t$ is identified, COLDB estimates a common preference vector $\overline{\boldsymbol{\theta}}_t$ for all users within this cluster by aggregating the historical feedback from all members of $\overline{C}_t$. Specifically, in Line 6, the common preference vector is determined by minimizing the following loss function:

$$\overline{\boldsymbol{\theta}}_t = \arg\min_{\boldsymbol{\theta}} - \sum_{\substack{s \in [t-1] \\ i_s \in \overline{C}_t}} \Big( y_s \log \mu \left( \boldsymbol{\theta}^\top [\phi(\boldsymbol{x}_{s,1}) - \phi(\boldsymbol{x}_{s,2})] \right)$$

$$+ (1 - y_s) \log \mu \left( \boldsymbol{\theta}^\top [\phi(\boldsymbol{x}_{s,2}) - \phi(\boldsymbol{x}_{s,1})] \right) \Big) + \frac{1}{2}\lambda \|\boldsymbol{\theta}\|_2^2 , \tag{1}$$

which corresponds to the Maximum Likelihood Estimation (MLE) using the data from all users in the cluster $\overline{C}_t$. Additionally, in Line 7, COLDB computes the aggregated information matrix for $\overline{C}_t$, which is subsequently utilized in selecting the second arm $\boldsymbol{x}_{t,2}$:

$$\boldsymbol{V}_{t-1} = \boldsymbol{V}_0 + \sum_{\substack{s \in [t-1] \\ i_s \in \overline{C}_t}} (\phi(\boldsymbol{x}_{s,1}) - \phi(\boldsymbol{x}_{s,2}))(\phi(\boldsymbol{x}_{s,1}) - \phi(\boldsymbol{x}_{s,2}))^\top \tag{2}$$

**Arm Recommendation Based on Cluster Statistics (Line 8-Line 9).** Leveraging the estimated common preference vector $\overline{\boldsymbol{\theta}}_t$ and the aggregated information matrix $\boldsymbol{V}_{t-1}$, COLDB proceeds to recommend two arms as follows:

- **First Arm Selection** ($\boldsymbol{x}_{t,1}$)**.** In Line 8, COLDB selects the first arm by greedily choosing the arm that maximizes the estimated reward according to $\overline{\boldsymbol{\theta}}_t$:

$$\boldsymbol{x}_{t,1} = \arg\max_{\boldsymbol{x} \in \mathcal{X}_t} \overline{\boldsymbol{\theta}}_t^\top \phi(\boldsymbol{x}). \tag{3}$$

- **Second Arm Selection** ($\boldsymbol{x}_{t,2}$)**.** Following the selection of $\boldsymbol{x}_{t,1}$, in Line 9, COLDB selects the second arm by maximizing an upper confidence bound (UCB):

$$\boldsymbol{x}_{t,2} = \arg\max_{\boldsymbol{x} \in \mathcal{X}_t} \overline{\boldsymbol{\theta}}_t^\top \phi(\boldsymbol{x}) + \frac{\beta_t}{\kappa_\mu} \|\phi(\boldsymbol{x}) - \phi(\boldsymbol{x}_{t,1})\|_{\boldsymbol{V}_{t-1}^{-1}} . \tag{4}$$

Intuitively, Eq.(4) encourages the selection of the arm which both (a) has a large predicted reward value and (b) is different from $\boldsymbol{x}_{t,1}$ and the arms selected in the previous $t - 1$ rounds when the served user belongs to the currently estimated cluster $\overline{C}_t$. In other words, the second arm $\boldsymbol{x}_{t,2}$ is chosen by balancing exploration and exploitation.

**Updating User Estimates and Interaction History (Line 10-Line 11).** Upon recommending $\boldsymbol{x}_{t,1}$ and $\boldsymbol{x}_{t,2}$, the user receives binary feedback $y_t = \mathbb{1}(\boldsymbol{x}_{t,1} \succ \boldsymbol{x}_{t,2})$ from user $i_t$, and then updates the interaction history $\mathcal{D}_t = \{i_s, \boldsymbol{x}_{s,1}, \boldsymbol{x}_{s,2}, y_s\}_{s=1}^t$ (Line 10). Moreover, COLDB updates the preference vector estimate for user $i_t$ while keeping the estimates for the other users unchanged (Line 11).

Specifically, the preference vector estimate $\hat{\boldsymbol{\theta}}_{i_t,t}$ is updated via MLE using the historical data from user $i_t$:

$$\hat{\boldsymbol{\theta}}_{i_t,t} = \arg\min_{\boldsymbol{\theta}} - \sum_{\substack{s \in [t-1] \\ i_s = i_t}} \Big( y_s \log \mu \big( \boldsymbol{\theta}^\top [\phi(\boldsymbol{x}_{s,1}) - \phi(\boldsymbol{x}_{s,2})] \big)$$

$$+ (1 - y_s) \log \mu \big( \boldsymbol{\theta}^\top [\phi(\boldsymbol{x}_{s,2}) - \phi(\boldsymbol{x}_{s,1})] \big) \Big) + \frac{\lambda}{2} \|\boldsymbol{\theta}\|_2^2 . \tag{5}$$

**Dynamic Graph Update (Line 12).** Finally, based on the updated preference estimate $\hat{\boldsymbol{\theta}}_{i_t,t}$ for user $i_t$, COLDB reassesses the similarity between $i_t$ and the other users. If the discrepancy between $\hat{\boldsymbol{\theta}}_{i_t,t}$ and $\hat{\boldsymbol{\theta}}_{\ell,t}$ for any user $\ell$ surpasses a predefined threshold (Line 12), the edge $(i_t, \ell)$ is removed from the graph $G_{t-1}$, effectively separating them into distinct clusters. The resultant graph $G_t = (\mathcal{U}, E_t)$ is then utilized in the subsequent rounds.

### 3.2. Clustering Of Neural Dueling Bandits (CONDB)

Our Clustering Of Neural Dueling Bandits (CONDB) algorithm is illustrated in Algorithm 2 (App. A), which adopts neural networks to model non-linear reward functions. Similar to COLDB, our CONDB algorithm also maintains a dynamic graph $G_t = (\mathcal{U}, E_t)$ in which every connected component denotes an inferred cluster, and adaptively removes the edges between users who are estimated to belong to different clusters.

**Cluster Inference $\overline{C}_t$ for User $i_t$ (Line 5).** Similar to COLDB (Algo. 1), when a new user $i_t$ arrives, our CONDB firstly identifies the connected component $\overline{C}_t$ in the maintained graph $G_{t-1}$ which contains the user $i_t$ and then uses it as the estimated cluster for $i_t$ (Line 5).

**Estimating Shared Statistics for Cluster $\overline{C}_t$ (Line 6).** After the cluster $\overline{C}_t$ is identified, our CONDB algorithm uses the history of preference feedback observations from all users in the cluster $\overline{C}_t$ to train a neural network (NN) to minimize the following loss function (Line 6):

$$\mathcal{L}_t(\boldsymbol{\theta}) = -\frac{1}{m} \sum_{\substack{s \in [t-1] \\ i_s \in \overline{C}_t}} \Big( y_s \log \mu \left( h(\boldsymbol{x}_{s,1}; \boldsymbol{\theta}) - h(\boldsymbol{x}_{s,2}; \boldsymbol{\theta}) \right) +$$

$$(1 - y_s) \log \mu \left( h(\boldsymbol{x}_{s,2}; \boldsymbol{\theta}) - h(\boldsymbol{x}_{s,1}; \boldsymbol{\theta}) \right) \Big) + \frac{\lambda}{2} \|\boldsymbol{\theta} - \boldsymbol{\theta}_0\|_2^2 \tag{9}$$

to yield parameters $\overline{\boldsymbol{\theta}}_t$. In addition, similar to COLDB (Algorithm 1), our CONDB computes the aggregated information matrix for the cluster $\overline{C}_t$ following Eq.(2). Note that here we replace $\phi(\boldsymbol{x})$ from Eq.(2) by the NTK feature representation $\phi(\boldsymbol{x}) = (1/\sqrt{m})g(\boldsymbol{x}; \boldsymbol{\theta}_0)$, in which $\boldsymbol{\theta}_0$ represents the initial parameters of the NN (Sec. 2.2).

**Arm Recommendation Based on Cluster Statistics (Line 8-Line 9).** Next, our CONDB algorithm leverages the

trained NN with parameters $\overline{\boldsymbol{\theta}}_t$ and the aggregated information matrix $\boldsymbol{V}_{t-1}$ to select the pair of arms. The first arm is selected by greedily maximizing the reward prediction of the NN with parameters $\overline{\boldsymbol{\theta}}_t$ (Line 8):

$$\boldsymbol{x}_{t,1} = \arg\max_{\boldsymbol{x} \in \mathcal{X}_t} h(\boldsymbol{x}; \overline{\boldsymbol{\theta}}_t). \tag{10}$$

The second arm is then selected optimistically (Line 9):

$$\boldsymbol{x}_{t,2} = \arg\max_{\boldsymbol{x} \in \mathcal{X}_t} h(\boldsymbol{x}; \overline{\boldsymbol{\theta}}_t) + \nu_T \left\| (\phi(\boldsymbol{x}) - \phi(\boldsymbol{x}_{t,1})) \right\|_{\boldsymbol{V}_{t-1}^{-1}}, \tag{11}$$

in which $\nu_T \triangleq \beta_T + B\sqrt{\frac{\lambda}{\kappa_\mu}} + 1$, $\beta_T \triangleq \frac{1}{\kappa_\mu} \sqrt{\widetilde{d} + 2\log(u/\delta)}$ and $B$ is defined in Lemma 2.6. Here $\widetilde{d}$ denotes the *effective dimenision* which we will introduce in detail in Sec. 4.2.

**Updating User Estimates and Interaction History (Line 10-Line 11).** After recommending the pair of arms $\boldsymbol{x}_{t,1}$ and $\boldsymbol{x}_{t,2}$, we collect the preference feedback $y_t = \mathbb{1}(\boldsymbol{x}_{t,1} \succ \boldsymbol{x}_{t,2})$ and update interaction history: $\mathcal{D}_t = \{i_s, \boldsymbol{x}_{s,1}, \boldsymbol{x}_{s,2}, y_s\}_{s=1}^t$ (Line 10). Next, we update the parameters of the NN used to predict the reward for user $i_t$ by minimizing the following loss function (Line 11):

$$\mathcal{L}_{i_t,t}(\boldsymbol{\theta}) = -\frac{1}{m_{\text{NN}}} \sum_{\substack{s \in [t-1] \\ i_s = i_t}} \Big( y_s \log \mu \left( h(\boldsymbol{x}_{s,1}; \boldsymbol{\theta}) - h(\boldsymbol{x}_{s,2}; \boldsymbol{\theta}) \right) +$$

$$(1 - y_s) \log \mu \left( h(\boldsymbol{x}_{s,2}; \boldsymbol{\theta}) - h(\boldsymbol{x}_{s,1}; \boldsymbol{\theta}) \right) \Big) + \frac{\lambda}{2} \|\boldsymbol{\theta} - \boldsymbol{\theta}_0\|_2^2 \tag{12}$$

to yield parameters $\hat{\boldsymbol{\theta}}_{i_t,t}$. The NN parameters for the other users remain unchanged.

**Dynamic Graph Update (Line 12).** Finally, we use the updated NN parameters $\hat{\boldsymbol{\theta}}_{i_t,t}$ for user $i_t$ to reassess the similarity between user $i_t$ and the other users. We remove the edge between $(i_t, \ell)$ from the graph $G_{t-1}$ if the difference between $\hat{\boldsymbol{\theta}}_{i_t,t}$ and $\hat{\boldsymbol{\theta}}_{\ell,t}$ is large enough (Line 12). Intuitively, if the estimated reward functions (represented by the respective parameters of their NNs for reward prediction) between two users are significantly different, we separate these two users into different clusters. The updated graph $G_t = (\mathcal{U}, E_t)$ is then used in the following rounds.

## 4. Theoretical Analysis

In this section, we present the theoretical results regarding the regret guarantees of our proposed algorithms and provide a detailed discussion of these findings.

### 4.1. Clustering Of Linear Dueling Bandits (COLDB)

The following theorem provides an upper bound on the expected regret achieved by the COLDB algorithm (Algo. 1) under the linear setting.

**Theorem 4.1.** *Suppose that Assumptions 2.1, 2.2, 2.3 and 2.4 are satisfied. Then the expected regret of the COLDB*

---

**Algorithm 1** Clustering Of Linear Dueling Bandits (COLDB)

---

1: **Input:** $f(T_{i,t}) = \frac{\sqrt{\lambda/\kappa_\mu} + \sqrt{2\log(u/\delta) + d\log(1 + 4T_{i,t}\kappa_\mu/d\lambda)}}{\kappa_\mu\sqrt{2\tilde{\lambda}_x T_{i,t}}}$, regularization parameter $\lambda > 0$, confidence parameter $\beta_t \triangleq \sqrt{2\log(1/\delta) + d\log(1 + tL^2\kappa_\mu/(d\lambda))}$, $\kappa_\mu > 0$.

2: **Initialization:** $V_0 = V_{i,0} = \frac{\lambda}{\kappa_\mu}I$, $\hat{\boldsymbol{\theta}}_{i,0} = \mathbf{0}$, $\forall i \in \mathcal{U}$, a complete Graph $G_0 = (\mathcal{U}, E_0)$ over $\mathcal{U}$.

3: **for** $t = 1, \ldots, T$ **do**

4:     Receive the index of the current user $i_t \in \mathcal{U}$, and the current feasible arm set $\mathcal{X}_t$;

5:     Find the connected component $\overline{C}_t$ for user $i_t$ in the current graph $G_{t-1}$ as the current cluster;

6:     Estimate the common preference vector $\overline{\boldsymbol{\theta}}_t$ for the current cluster $\overline{C}_t$:

$$\overline{\boldsymbol{\theta}}_t = \arg\min_{\boldsymbol{\theta}} - \sum_{\substack{s \in [t-1] \\ i_s \in \overline{C}_t}} \left( y_s \log \mu \left( \boldsymbol{\theta}^\top [\phi(\boldsymbol{x}_{s,1}) - \phi(\boldsymbol{x}_{s,2})] \right) + (1 - y_s) \log \mu \left( \boldsymbol{\theta}^\top [\phi(\boldsymbol{x}_{s,2}) - \phi(\boldsymbol{x}_{s,1})] \right) \right) + \frac{\lambda}{2} \|\boldsymbol{\theta}\|_2^2 ; \tag{6}$$

7:     Calculate aggregated information matrix for cluster $\overline{C}_t$: $V_{t-1} = V_0 + \sum_{\substack{s \in [t-1] \\ i_s \in \overline{C}_t}} (\phi(\boldsymbol{x}_{s,1}) - \phi(\boldsymbol{x}_{s,2}))(\phi(\boldsymbol{x}_{s,1}) - \phi(\boldsymbol{x}_{s,2}))^\top$.

8:     Choose the first arm $\boldsymbol{x}_{t,1} = \arg\max_{\boldsymbol{x} \in \mathcal{X}_t} \overline{\boldsymbol{\theta}}_t^\top \phi(\boldsymbol{x})$;

9:     Choose the second arm $\boldsymbol{x}_{t,2} = \arg\max_{\boldsymbol{x} \in \mathcal{X}_t} \overline{\boldsymbol{\theta}}_t^\top (\phi(\boldsymbol{x}) - \phi(\boldsymbol{x}_{t,1})) + \frac{\beta_t}{\kappa_\mu} \|\phi(\boldsymbol{x}) - \phi(\boldsymbol{x}_{t,1})\|_{V_{t-1}^{-1}}$;

10:     Observe the preference feedback: $y_t = \mathbb{1}(\boldsymbol{x}_{t,1} \succ \boldsymbol{x}_{t,2})$, and update history: $\mathcal{D}_t = \{i_s, \boldsymbol{x}_{s,1}, \boldsymbol{x}_{s,2}, y_s\}_{s=1,\ldots,t}$;

11:     Update the estimation for the current served user $i_t$:

$$\hat{\boldsymbol{\theta}}_{i_t,t} = \arg\min_{\boldsymbol{\theta}} - \sum_{\substack{s \in [t-1] \\ i_s = i_t}} \left( y_s \log \mu \left( \boldsymbol{\theta}^\top [\phi(\boldsymbol{x}_{s,1}) - \phi(\boldsymbol{x}_{s,2})] \right) + (1 - y_s) \log \mu \left( \boldsymbol{\theta}^\top [\phi(\boldsymbol{x}_{s,2}) - \phi(\boldsymbol{x}_{s,1})] \right) \right) + \frac{\lambda}{2} \|\boldsymbol{\theta}\|_2^2, \tag{7}$$

        keep the estimations of other users unchanged;

12:     Delete the edge $(i_t, \ell) \in E_{t-1}$ if

$$\left\| \hat{\boldsymbol{\theta}}_{i_t,t} - \hat{\boldsymbol{\theta}}_{\ell,t} \right\|_2 > f(T_{i_t,t}) + f(T_{\ell,t}) \tag{8}$$

13: **end for**

---

*algorithm (Algo. 1) for $T$ rounds satisfies*

$$R(T) = O\left(u\left(\frac{d}{\kappa_\mu^2 \tilde{\lambda}_x \gamma^2} + \frac{1}{\tilde{\lambda}_x^2}\right) \log T + \frac{1}{\kappa_\mu} d\sqrt{mT}\right) \tag{13}$$

$$= O\left(\frac{1}{\kappa_\mu} d\sqrt{mT}\right), \tag{14}$$

*where $\tilde{\lambda}_x \triangleq \int_0^{\lambda_x} (1 - e^{-\frac{(\lambda_x - x)^2}{2\sigma^2}})^C dx$ is the problem instance dependent constant (Wang et al., 2024a;b).*

The proof of this theorem can be found in Appendix B. The regret bound in Eq.(13) consists of two terms. The first term accounts for the number of rounds required to accumulate sufficient information to correctly cluster all users with high probability, and it scales only logarithmically with the number of time steps $T$. The second term captures the regret after successfully clustering the users, which depends on the number of clusters $m$, rather than the potentially huge total number of users $u$. Notably, the regret upper bound is not

only sub-linear in $T$, but also *becomes tighter when there is a smaller number of clusters $m$*, i.e., when a larger number of users belong to the same cluster on average. This provides a formal justification for the advantage of cross-user collaboration in our problem setting where only preference feedback is available.

Based on prior techniques (Wang et al., 2024a; Liu et al., 2022) and the single-user dueling bandit lower bound (Saha, 2021), we can get a lower bound of $O(\sqrt{dmT})$ for the linear setting. Our Algo.1 achieves an upper bound of $O(d\sqrt{mT})$, which is tight up to a $\sqrt{d}$ factor—a common gap in linear bandits (e.g., LinUCB). Thus, our upper bound is tight and optimal in $m$ for the linear case.

In the special case where there is only one user ($m = 1$), the regret bound simplifies to $O(d\sqrt{T}/\kappa_\mu)$, which aligns with the classical results in the single-user linear dueling bandit literature (Saha, 2021; Bengs et al., 2022; Li et al., 2024). Compared to the previous works on clustering of bandits

with linear reward functions (Gentile et al., 2014; Wang et al., 2024a; Li et al., 2019), our regret upper bound has an extra dependency on $1/\kappa_\mu$. Since $\kappa_\mu < 0.25$ for the logistic function, this dependency makes our regret upper bound larger and hence captures the more challenging nature of the preference feedback compared to the numerical feedback in classical clustering of linear bandits.

## 4.2. Clustering Of Neural Dueling Bandits (CONDB)

Let $\mathbf{H}' = \sum_{t=1}^{T} \sum_{(i,j)\in C_K^2} z_j^i(t) z_j^i(t)^\top \frac{1}{m_{\mathrm{NN}}}$, in which $z_j^i(t) = g(\boldsymbol{x}_{t,i}; \boldsymbol{\theta}_0) - g(\boldsymbol{x}_{t,j}; \boldsymbol{\theta}_0)$ and $C_K^2$ denotes all pairwise combinations of $K$ arms. Then, the effective dimension $\widetilde{d}$ is defined as follows (Verma et al., 2024):

$$\widetilde{d} = \log \det \left( \frac{\kappa_\mu}{\lambda} \mathbf{H}' + \mathbf{I} \right). \tag{15}$$

The definition of $\widetilde{d}$ considers the contexts from all users and in all $T$ rounds. The theorem below gives an upper bound on the expected regret of our CONDB algorithm (Algo. 2).

**Theorem 4.2.** *Suppose that Assumptions 2.1, 2.4, 2.5 and 2.7 are satisfied (let $\phi(\boldsymbol{x}) = (1/\sqrt{m_{NN}})g(\boldsymbol{x}; \boldsymbol{\theta}_0)$ in Assumption 2.4). As long as $m_{NN} \geq poly(T, L, K, 1/\kappa_\mu, L_\mu, 1/\lambda_0, 1/\lambda, \log(1/\delta))$, then the expected regret of the CONDB algorithm (Algo. 2) for $T$ rounds satisfies*

$$R_T = O\left( u\big( \frac{\widetilde{d}}{\kappa_\mu^2 \widetilde{\lambda}_x \gamma^2} + \frac{1}{\widetilde{\lambda}_x^2} \big) \log T + \right.$$
$$\left. \big( \frac{\sqrt{\widetilde{d}}}{\kappa_\mu} + B\sqrt{\frac{\lambda}{\kappa_\mu}} \big) \sqrt{\widetilde{d} m T} \right) \tag{16}$$

$$= O\left( \big( \frac{\sqrt{\widetilde{d}}}{\kappa_\mu} + B\sqrt{\frac{\lambda}{\kappa_\mu}} \big) \sqrt{\widetilde{d} m T} \right). \tag{17}$$

The proof of this theorem can be found in Appendix C. The first term in the regret bound in Eq. 16 has the same form as the first term in the regret bound of COLDB in Eq.(13), except that the input dimension $d$ for COLDB (Eq.(13)) is replaced by the effective dimension $\widetilde{d}$ for CONDB (Eq.(16)). As discussed in Verma et al. (2024), $\widetilde{d}$ is usually larger than the effective dimension in classical neural bandits (Zhou et al., 2020; Zhang et al., 2021). This dependency, together with the extra dependency on $1/\kappa_\mu$, reflects the added difficulty from the preference feedback compared to the more informative numerical feedback in classical neural bandits.

Similar to COLDB (Theorem 4.1), the first term in the regret upper bound of CONDB (Theorem 4.2) results from the number of rounds needed to collect enough observations to correctly identify the clustering structure. The second term corresponds to the regret of all users after the correct clustering structure is identified, which depends on the number of clusters $m$ instead of the number of users $u$. Theorem

4.2 also shows that the regret upper bound of CONDB is sub-linear in $T$, and becomes improved as the number of users belonging to the same cluster is increased on average (i.e., when the number of clusters $m$ is smaller). Moreover, in the special case where the number of clusters is $m = 1$, the regret upper bound in Eq.(17) becomes the same as that of the standard neural dueling bandits (Verma et al., 2024).

## 5. Experimental Results

We use both synthetic and real-world experiments to evaluate the performance of our COLDB and CONDB algorithms. For both algorithms, we compare them with their corresponding single-user variant as the baseline. Specifically, for COLDB, we compare it with the baseline of LDB_IND, which refers to Linear Dueling Bandit (Independent) (Bengs et al., 2022), meaning running independent classic linear dueling bandit algorithms for each user separately; similarly, for CONDB, we compare it with NDB_IND, which stands for Neural Dueling Bandit (Independent) (Verma et al., 2024).

**COLDB.** Our experimental settings mostly follow the designs from the works on clustering of bandits (Wang et al., 2024a; Li et al., 2019). In our synthetic experiment for COLDB, we design a setting with linear reward functions: $f_i(\boldsymbol{x}) = \boldsymbol{\theta}_i^\top \boldsymbol{x}$. We choose $u = 200$ users, $K = 20$ arms and a feature dimension of $d = 20$, and construct two settings with $m = 2$ and $m = 5$ groundtruth clusters, respectively. In the experiment with the MovieLens dataset (Harper & Konstan, 2015), we follow the experimental setting from Wang et al. (2024a), a setting with 200 users. Same as the synthetic experiment, we choose the number of arms in every round to be $K = 20$ and let the input feature dimension be $d = 20$. We construct a setting with $m = 5$ clusters. We repeat each experiment for three independent trials and report the mean ± standard error.

Fig. 1 plots the cumulative regret of our COLDB and the baseline of LDB_IND. The results show that our COLDB algorithm significantly outperforms the baseline of LDB_IND in both the synthetic and real-world experiments. Moreover, Fig. 1 (a) demonstrates that when $m = 2$ (i.e., when a larger number of users belong to the same cluster on average), the performance of our COLDB is improved, which is *consistent with our theoretical results* (Sec. 4.1).

**CONDB.** We also construct both a synthetic and real-world experiment to evaluate our CONDB algorithm. Most of the experimental settings are the same as those of the COLDB algorithm described above. The major difference is that instead of using linear reward functions, here we adopt a non-linear reward function, i.e., a square function: $f_i(\boldsymbol{x}) = (\boldsymbol{\theta}_i^\top \boldsymbol{x})^2$. The results in this setting are plotted

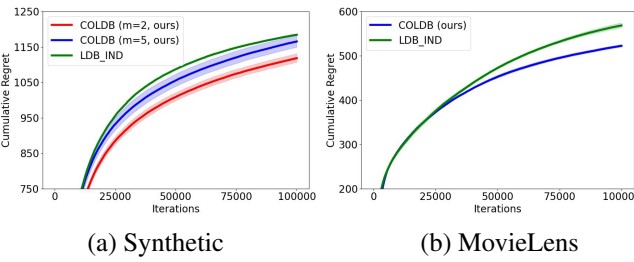

(a) Synthetic       (b) MovieLens

*Figure 1.* Experimental results for our COLDB algorithm with a linear reward function.

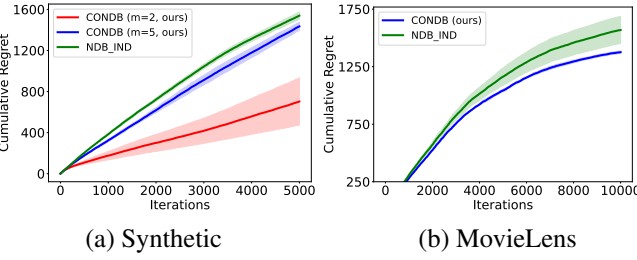

(a) Synthetic       (b) MovieLens

*Figure 2.* Experimental results for our CONDB algorithm with a non-linear (square) reward function.

in Fig. 2. Our CONDB algorithm achieves significantly smaller cumulative regrets than the baseline algorithm of NDB_IND in both the synthetic and real-world experiments. Moreover, Fig. 2 (a) shows that the performance of our CONDB is improved when a larger number of users are in the same cluster on average, i.e., when $m = 2$. These results demonstrate the potential of our CONDB algorithm to excel in problems with complicated non-linear reward functions.

# 6. Related Work

Our work is closely related to: online clustering of bandits (CB), dueling bandits, and neural bandits.

## 6.1. Clustering of Bandits

The concept of clustering bandits (CB) was first introduced in (Gentile et al., 2014), where a graph-based approach was proposed for solving the problem. In subsequent work, (Li et al., 2016) explored the incorporation of collaborative effects among items to aid in the clustering of users. Further extending this idea, (Li & Zhang, 2018) tackled the CB problem in the context of cascading bandits, where feedback is provided through random prefixes. Another direction of this research, presented in (Li et al., 2019), investigates the scenario where users have varying arrival frequencies. In (Liu et al., 2022; Yang et al., 2024), federated settings for CB are proposed, which addresses both privacy concerns and the communication overhead in distributed environments. More

recently, papers by (Wang et al., 2024a; Dai et al., 2024a) and (Wang et al., 2024b; Dai et al., 2022) examine the design of robust CB algorithms in the presence of model mis-specifications and adversarial data corruptions, respectively.

All these works in CB assume the agent recommends a single arm per round, with a real-valued reward reflecting user satisfaction. However, this does not apply to scenarios such as large language models seeking user preference feedback to improve the model, where users provide binary feedback comparing two responses. To the best of our knowledge, this paper is the first to consider dueling binary feedback in the CB problem.

## 6.2. Dueling Bandits and Neural Bandits

Dueling bandits has been receiving growing attention over the years since its introduction (Yue & Joachims, 2009; 2011; Yue et al., 2012) due to the prevalence of preference or relative feedback in real-world applications. Many earlier works on dueling bandits have focused on MAB problems with a finte number of arms (Zoghi et al., 2014b; Ailon et al., 2014; Zoghi et al., 2014a; Komiyama et al., 2015; Gajane et al., 2015; Saha & Gopalan, 2018; 2019a;b; Saha & Ghoshal, 2022; Zhu et al., 2023). More recently, con-textual dueing bandits, which model the reward function using a parametric function of the features of the arms, have attracted considerable attention (Saha, 2021; Saha & Krish-namurthy, 2022; Bengs et al., 2022; Di et al., 2023; Li et al., 2024; Verma et al., 2024).

To apply MABs to complicated real-world applications with non-linear reward functions, neural bandits have been proposed which use a neural network to model the reward function (Zhou et al., 2020; Zhang et al., 2021). Recently, we have witnessed a significant growing interest in further improving the theoretical and empirical performance of neural bandits and applying it to solve real-world problems (Xu et al., 2020; Kassraie & Krause, 2022; Gu et al., 2021; Nabati et al., 2021; Lisicki et al., 2021; Ban et al., 2022; Ban & He, 2021a; Jia et al., 2021; Nguyen-Tang et al., 2022; Zhu et al., 2021; Kassraie et al., 2022; Salgia et al., 2022; Dai et al., 2022; Hwang et al., 2023; Qi et al., 2023; 2024). In particular, the work of Ban et al. (2024) has adopted a neural network as a meta-learner for adapting to users in different clusters within the framework of clustering of bandits, and the work of Verma et al. (2024) has combined neural bandits with dueling bandits.

# 7. Conclusion

In this work, we introduce the first clustering of dueling bandit algorithms for both linear and non-linear latent re-ward functions, which enhance the performance of MAB with preference feedback via cross-user collaboraiton. Our

algorithms estimates the clustering structure online based on the estimated reward function parameters, and employs the data from all users within the same cluster to select the pair of arms to query for preference feedback. We derive upper bounds on the cumulative regret of our algorithms, which show that our algorithms enjoy theoretically guaranteed improvement when a larger number of users belong to the same cluster on average. We also use synthetic and real-world experiments to validate our theoretical findings.

## Acknowledgement

The work of Qinghua Hu is supported by the National Natural Science Foundation of China (Key Program under Grant No. U23B2049).

## Impact Statement

This paper presents work whose goal is to advance the field of Machine Learning. There are many potential societal consequences of our work, none which we feel must be specifically highlighted here.

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

# A. Clustering Of Neural Dueling Bandits (CONDB) Algorithm

Here we provide the complete statement of our CONDB algorithm.

---

**Algorithm 2** Clustering Of Neural Dueling Bandits (CONDB)

---

1: **Input:** $f(T_{i,t}) \triangleq \frac{\beta_T + B\sqrt{\frac{\lambda}{\kappa_\mu}} + 1}{\sqrt{2\tilde{\lambda}_x T_{i,t}}}$, regularization parameter $\lambda > 0$, confidence parameter $\beta_T \triangleq \frac{1}{\kappa_\mu}\sqrt{\tilde{d} + 2\log(u/\delta)}$.
   $\phi(\boldsymbol{x}) = \frac{1}{\sqrt{m_{\text{NN}}}} g(\boldsymbol{x}; \boldsymbol{\theta}_0)$ where $\boldsymbol{\theta}_0$ denotes the NN parameters at initialization.

2: **Initialization:** $\boldsymbol{V}_0 = \boldsymbol{V}_{i,0} = \frac{\lambda}{\kappa_\mu} \mathbf{I}$, $\hat{\boldsymbol{\theta}}_{i,0} = \boldsymbol{0}$, $\forall i \in \mathcal{U}$, a complete Graph $G_0 = (\mathcal{U}, E_0)$ over $\mathcal{U}$.

3: **for** $t = 1, \ldots, T$ **do**

4:   Receive the index of the current user $i_t \in \mathcal{U}$, and the current feasible arm set $\mathcal{X}_t$;

5:   Find the connected component $\overline{C}_t$ for user $i_t$ in the current graph $G_{t-1}$ as the current cluster;

6:   Train the neural network using $\{(\boldsymbol{x}_{s,1}, \boldsymbol{x}_{s,2}, y_s)\}_{s\in[t-1], i_s\in\overline{C}_t}$ by minimizing the following loss function:

$$\overline{\boldsymbol{\theta}}_t = \arg\min_{\boldsymbol{\theta}} -\frac{1}{m} \sum_{\substack{s\in[t-1]\\ i_s\in\overline{C}_t}} (y_s \log\mu\left(h(\boldsymbol{x}_{s,1}; \boldsymbol{\theta}) - h(\boldsymbol{x}_{s,2}; \boldsymbol{\theta})\right) + (1-y_s)\log\mu\left(h(\boldsymbol{x}_{s,2}; \boldsymbol{\theta}) - h(\boldsymbol{x}_{s,1}; \boldsymbol{\theta})\right)) + $$
$$\frac{\lambda}{2}\|\boldsymbol{\theta} - \boldsymbol{\theta}_0\|_2^2; \tag{18}$$

7:   Calculate the aggregated information matrix for cluster $\overline{C}_t$: $\boldsymbol{V}_{-1} = \boldsymbol{V}_0 + \sum_{\substack{s\in[t-1]\\ i_s\in\overline{C}_t}} (\phi(\boldsymbol{x}_{s,1}) - \phi(\boldsymbol{x}_{s,2}))(\phi(\boldsymbol{x}_{s,1}) - \phi(\boldsymbol{x}_{s,2}))^\top$.

8:   Choose the first arm $\boldsymbol{x}_{t,1} = \arg\max_{\boldsymbol{x}\in\mathcal{X}_t} h(\boldsymbol{x}; \overline{\boldsymbol{\theta}}_t)$;

9:   Choose the second arm $\boldsymbol{x}_{t,2} = \arg\max_{\boldsymbol{x}\in\mathcal{X}_t} h(\boldsymbol{x}; \overline{\boldsymbol{\theta}}_t) + \left(\beta_T + B\sqrt{\frac{\lambda}{\kappa_\mu}} + 1\right)\|(\phi(\boldsymbol{x}) - \phi(\boldsymbol{x}_{t,1}))\|_{\boldsymbol{V}_{t-1}^{-1}}$;

10:   Observe the preference feedback: $y_t = \mathbb{1}(\boldsymbol{x}_{t,1} \succ \boldsymbol{x}_{t,2})$, and update history: $\mathcal{D}_t = \{i_s, \boldsymbol{x}_{s,1}, \boldsymbol{x}_{s,2}, y_s\}_{s=1,\ldots,t}$;

11:   Train the neural network using all data for user $i_t$: $\{(\boldsymbol{x}_{s,1}, \boldsymbol{x}_{s,2}, y_s)\}_{s\in[t], i_s=i_t}$ by minimizing the following loss function:

$$\hat{\boldsymbol{\theta}}_{i_t,t} = \arg\min_{\boldsymbol{\theta}} -\frac{1}{m_{\text{NN}}} \sum_{\substack{s\in[t-1]\\ i_s=i_t}} (y_s \log\mu\left(h(\boldsymbol{x}_{s,1}; \boldsymbol{\theta}) - h(\boldsymbol{x}_{s,2}; \boldsymbol{\theta})\right) + (1-y_s)\log\mu\left(h(\boldsymbol{x}_{s,2}; \boldsymbol{\theta}) - h(\boldsymbol{x}_{s,1}; \boldsymbol{\theta})\right)) + $$
$$\frac{\lambda}{2}\|\boldsymbol{\theta} - \boldsymbol{\theta}_0\|_2^2; \tag{19}$$

   keep the estimations of other users unchanged;

12:   Delete the edge $(i_t, \ell) \in E_{t-1}$ if

$$\sqrt{m_{\text{NN}}}\left\|\hat{\boldsymbol{\theta}}_{i_t,t} - \hat{\boldsymbol{\theta}}_{\ell,t}\right\|_2 > f(T_{i_t,t}) + f(T_{\ell,t}) \tag{20}$$

13: **end for**

---

# B. Proof of Theorem 4.1

First, we prove the following lemma.

**Lemma B.1.** *With probability at least $1 - \delta$ for some $\delta \in (0, 1)$, at any $t \in [T]$:*

$$\left\|\hat{\boldsymbol{\theta}}_{i,t} - \boldsymbol{\theta}^{j(i)}\right\|_2 \leq \frac{\sqrt{\lambda\kappa_\mu} + \sqrt{2\log(u/\delta) + d\log(1 + T_{i,t}\kappa_\mu/d\lambda)}}{\kappa_\mu\sqrt{\lambda_{min}(\boldsymbol{V}_{i,t-1})}}, \forall i \in \mathcal{U}, \tag{21}$$

*where $\boldsymbol{V}_{i,t-1} = \frac{\lambda}{\kappa_\mu}\mathbf{I} + \sum_{\substack{s\in[t-1]\\ i_s=i}} (\phi(\boldsymbol{x}_{s,1}) - \phi(\boldsymbol{x}_{s,2}))(\phi(\boldsymbol{x}_{s,1}) - \phi(\boldsymbol{x}_{s,2}))^\top$, and $T_{i,t}$ denotes the number of rounds of seeing user $i$ in the first $t$ rounds.*

*Proof.* First, we prove the following result.

For a fixed user $i$, with probability at least $1 - \delta$ for some $\delta \in (0, 1)$, at any $t \in [T]$:

$$\left\| \hat{\boldsymbol{\theta}}_{i,t} - \boldsymbol{\theta}^{j(i)} \right\|_{\boldsymbol{V}_{i,t-1}} \leq \frac{\sqrt{\lambda \kappa_\mu} + \sqrt{2 \log(1/\delta) + d \log(1 + 4T_{i,t} \kappa_\mu / d\lambda)}}{\kappa_\mu}, \tag{22}$$

Recall that $f_i(\boldsymbol{x}) = \boldsymbol{\theta}_i^\top \phi(\boldsymbol{x})$. In iteration $s$, define $\widetilde{\phi}_s = \phi(\boldsymbol{x}_{s,1}) - \phi(\boldsymbol{x}_{s,2})$. And we define $\widetilde{f}_{i,s} = f_i(\boldsymbol{x}_{s,1}) - f_i(\boldsymbol{x}_{s,2}) = \boldsymbol{\theta}_i^\top \widetilde{\phi}_s$.

For any $\boldsymbol{\theta}_{f'} \in \mathbb{R}^d$, define

$$G_{i,t}(\boldsymbol{\theta}_{f'}) = \sum_{\substack{s \in [t-1]: \\ i_s = i}} \left( \mu(\boldsymbol{\theta}_{f'}^\top \widetilde{\phi}_s) - \mu(\boldsymbol{\theta}_i^\top \widetilde{\phi}_s) \right) \widetilde{\phi}_s + \lambda \boldsymbol{\theta}_{f'}.$$

For $\lambda' \in (0, 1)$, setting $\boldsymbol{\theta}_{\bar{f}} = \lambda' \boldsymbol{\theta}_{f_1'} + (1 - \lambda') \boldsymbol{\theta}_{f_2'}$. and using the mean-value theorem, we get:

$$G_{i,t}(\boldsymbol{\theta}_{f_1'}) - G_{i,t}(\boldsymbol{\theta}_{f_2'}) = \left[ \sum_{\substack{s \in [t-1]: \\ i_s = i}} \nabla \mu(\boldsymbol{\theta}_{\bar{f}}^\top \widetilde{\phi}_s) \widetilde{\phi}_s \widetilde{\phi}_s^\top + \lambda \mathbf{I} \right] (\boldsymbol{\theta}_{f_1'} - \boldsymbol{\theta}_{f_2'}) \qquad (\boldsymbol{\theta}_i \text{ is constant}) \tag{23}$$

Define $\boldsymbol{M}_{i,t-1} = \left[ \sum_{\substack{s \in [t-1]: \\ i_s = i}} \nabla \mu(\boldsymbol{\theta}_{\bar{f}}^\top \widetilde{\phi}_s) \widetilde{\phi}_s \widetilde{\phi}_s^\top + \lambda \mathbf{I} \right]$, and recall that $\boldsymbol{V}_{i,t-1} = \sum_{\substack{s \in [t-1]: \\ i_s = i}} \widetilde{\phi}_s \widetilde{\phi}_s^\top + \frac{\lambda}{\kappa_\mu} \mathbf{I}$. Then we have that $\boldsymbol{M}_{i,t-1} \succeq \kappa_\mu \boldsymbol{V}_{i,t-1}$ and that $\boldsymbol{V}_{i,t-1}^{-1} \succeq \kappa_\mu \boldsymbol{M}_{i,t-1}^{-1}$, where we use the notation $\boldsymbol{M} \succeq \boldsymbol{V}$ to denote that $\boldsymbol{M} - \boldsymbol{V}$ is a positive semi-definite matrix. Then we have

$$
\begin{aligned}
\left\| G_{i,t}(\hat{\boldsymbol{\theta}}_{i,t}) - \lambda \boldsymbol{\theta}_i \right\|_{\boldsymbol{V}_{i,t-1}^{-1}}^2 &= \left\| G_{i,t}(\boldsymbol{\theta}_i) - G_t(\hat{\boldsymbol{\theta}}_{i,t}) \right\|_{\boldsymbol{V}_{i,t-1}^{-1}}^2 = \left\| \boldsymbol{M}_{i,t-1}(\boldsymbol{\theta}_i - \hat{\boldsymbol{\theta}}_{i,t}) \right\|_{\boldsymbol{V}_{i,t-1}^{-1}}^2 \quad (G_{i,t}(\boldsymbol{\theta}_i) = \lambda \boldsymbol{\theta}_i \text{ by definition}) \\
&= (\boldsymbol{\theta}_i - \hat{\boldsymbol{\theta}}_{i,t})^\top \boldsymbol{M}_{i,t-1} \boldsymbol{V}_{i,t-1}^{-1} \boldsymbol{M}_{i,t-1}(\boldsymbol{\theta}_i - \hat{\boldsymbol{\theta}}_{i,t}) \\
&\geq (\boldsymbol{\theta}_i - \hat{\boldsymbol{\theta}}_{i,t})^\top \boldsymbol{M}_{i,t-1} \kappa_\mu \boldsymbol{M}_{i,t-1}^{-1} \boldsymbol{M}_{i,t-1}(\boldsymbol{\theta}_i - \hat{\boldsymbol{\theta}}_{i,t}) \\
&= \kappa_\mu (\boldsymbol{\theta}_i - \hat{\boldsymbol{\theta}}_{i,t})^\top \boldsymbol{M}_{i,t-1}(\boldsymbol{\theta}_i - \hat{\boldsymbol{\theta}}_{i,t}) \\
&\geq \kappa_\mu (\boldsymbol{\theta}_i - \hat{\boldsymbol{\theta}}_{i,t})^\top \kappa_\mu \boldsymbol{V}_{i,t-1}(\boldsymbol{\theta}_i - \hat{\boldsymbol{\theta}}_{i,t}) \\
&= \kappa_\mu^2 (\boldsymbol{\theta}_i - \hat{\boldsymbol{\theta}}_{i,t})^\top \boldsymbol{V}_{i,t-1}(\boldsymbol{\theta}_i - \hat{\boldsymbol{\theta}}_{i,t}) \\
&= \kappa_\mu^2 \left\| \boldsymbol{\theta}_i - \hat{\boldsymbol{\theta}}_{i,t} \right\|_{\boldsymbol{V}_{i,t-1}}^2 \qquad (\text{as } \|\boldsymbol{x}\|_A^2 = \boldsymbol{x}^\top \boldsymbol{A} \boldsymbol{x})
\end{aligned}
$$

The first inequality is because $\boldsymbol{V}_{i,t-1}^{-1} \succeq \kappa_\mu \boldsymbol{M}_{i,t-1}^{-1}$, and the second inequality follows from $\boldsymbol{M}_{i,t-1} \succeq \kappa_\mu \boldsymbol{V}_{i,t-1}$.

Note that $\frac{\kappa_\mu}{\lambda} \mathbf{I} \succeq \boldsymbol{V}_{i,t-1}$, which allows us to show that

$$\|\lambda \boldsymbol{\theta}_i\|_{\boldsymbol{V}_{i,t-1}^{-1}} = \lambda \sqrt{\boldsymbol{\theta}_i^\top \boldsymbol{V}_{i,t-1}^{-1} \boldsymbol{\theta}_i} \leq \lambda \sqrt{\boldsymbol{\theta}_i^\top \frac{\kappa_\mu}{\lambda} \boldsymbol{\theta}_i} \leq \sqrt{\lambda \kappa_\mu} \|\boldsymbol{\theta}_i\|_2 \leq \sqrt{\lambda \kappa_\mu}. \tag{24}$$

Using the two equations above, we have that

$$
\begin{aligned}
\left\| \boldsymbol{\theta}_i - \hat{\boldsymbol{\theta}}_{i,t} \right\|_{\boldsymbol{V}_{i,t-1}} &\leq \frac{1}{\kappa_\mu} \left\| G_{i,t}(\hat{\boldsymbol{\theta}}_{i,t}) - \lambda \boldsymbol{\theta}_i \right\|_{\boldsymbol{V}_{i,t-1}^{-1}} \leq \frac{1}{\kappa_\mu} \left\| G_{i,t}(\hat{\boldsymbol{\theta}}_{i,t}) \right\|_{\boldsymbol{V}_{i,t-1}^{-1}} + \frac{1}{\kappa_\mu} \|\lambda \boldsymbol{\theta}_i\|_{\boldsymbol{V}_{i,t-1}^{-1}} \\
&\leq \frac{1}{\kappa_\mu} \left\| G_{i,t}(\hat{\boldsymbol{\theta}}_{i,t}) \right\|_{\boldsymbol{V}_{i,t-1}^{-1}} + \sqrt{\frac{\lambda}{\kappa_\mu}}
\end{aligned} \tag{25}
$$

Then, let $f_{t,s}^i = \hat{\boldsymbol{\theta}}_{i,t}^\top \tilde{\phi}_s$, we have:

$$
\frac{1}{\kappa_\mu^2} \left\| G_{i,t}(\hat{\boldsymbol{\theta}}_{i,t}) \right\|_{\boldsymbol{V}_{i,t-1}^{-1}}^2 = \frac{1}{\kappa_\mu^2} \left\| \sum_{\substack{s\in[t-1]:\\i_s=i}} (\mu(\hat{\boldsymbol{\theta}}_{i,t}^\top \tilde{\phi}_s) - \mu(\boldsymbol{\theta}_i^\top \tilde{\phi}_s))\tilde{\phi}_s + \lambda\hat{\boldsymbol{\theta}}_{i,t} \right\|_{\boldsymbol{V}_{i,t-1}^{-1}}^2 \qquad \left( \text{by definition of } G_{i,t}(\hat{\boldsymbol{\theta}}_{i,t}) \right)
$$

$$
= \frac{1}{\kappa_\mu^2} \left\| \sum_{\substack{s\in[t-1]:\\i_s=i}} (\mu(f_{t,s}^i) - \mu(\widetilde{f}_{i,s}))\tilde{\phi}_s + \lambda\hat{\boldsymbol{\theta}}_{i,t} \right\|_{\boldsymbol{V}_{i,t-1}^{-1}}^2 \qquad \left( \text{see definitions of } f_{t,s}^i \text{ and } \widetilde{f}_{i,s} \right)
$$

$$
= \frac{1}{\kappa_\mu^2} \left\| \sum_{\substack{s\in[t-1]:\\i_s=i}} (\mu(f_{t,s}^i) - (y_s - \epsilon_s))\tilde{\phi}_s + \lambda\hat{\boldsymbol{\theta}}_{i,t} \right\|_{\boldsymbol{V}_{i,t-1}^{-1}}^2 \qquad \left( \text{as } y_s = \mu(\widetilde{f}_{i,s}) + \epsilon_s \text{if } i_s = i \right)
$$

$$
= \frac{1}{\kappa_\mu^2} \left\| \sum_{\substack{s\in[t-1]:\\i_s=i}} \left(\mu(f_{t,s}^i) - y_s\right)\tilde{\phi}_s + \sum_{\substack{s\in[t-1]:\\i_s=i}} \epsilon_s\tilde{\phi}_s + \lambda\hat{\boldsymbol{\theta}}_{i,t} \right\|_{\boldsymbol{V}_{i,t-1}^{-1}}^2
$$

$$
\leq \frac{1}{\kappa_\mu^2} \left\| \sum_{\substack{s\in[t-1]:\\i_s=i}} \epsilon_s\tilde{\phi}_s \right\|_{\boldsymbol{V}_{i,t-1}^{-1}}^2 .
$$

The last step holds due to the following reasoning. Recall that $\hat{\boldsymbol{\theta}}_{i,t}$ is computed using MLE by solving the following equation:

$$
\hat{\boldsymbol{\theta}}_{i_t,t} = \arg\min_{\boldsymbol{\theta}} \left[ -\sum_{\substack{s\in[t-1]\\i_s=i_t}} \left( y_s \log\mu\big(\boldsymbol{\theta}^\top[\phi(\boldsymbol{x}_{s,1})-\phi(\boldsymbol{x}_{s,2})]\big) + (1-y_s)\log\mu\big(\boldsymbol{\theta}^\top[\phi(\boldsymbol{x}_{s,2})-\phi(\boldsymbol{x}_{s,1})]\big) \right) + \frac{\lambda}{2}\|\boldsymbol{\theta}\|_2^2 \right]. \quad (26)
$$

Setting its gradient to 0, the following is satisfied:

$$
\sum_{\substack{s\in[t-1]:\\i_s=i}} \left( \mu\left(\hat{\boldsymbol{\theta}}_{i,t}^\top\tilde{\phi}_s\right) - y_s \right)\tilde{\phi}_s + \lambda\hat{\boldsymbol{\theta}}_{i,t} = 0, \qquad (27)
$$

which is used in the last step.

Now we have

$$
\frac{1}{\kappa_\mu^2} \left\| G_{i,t}(\hat{\boldsymbol{\theta}}_{i,t}) \right\|_{\boldsymbol{V}_{i,t-1}^{-1}}^2 \leq \frac{1}{\kappa_\mu^2} \left\| \sum_{\substack{s\in[t-1]:\\i_s=i}} \epsilon_s\tilde{\phi}_s \right\|_{\boldsymbol{V}_{i,t-1}^{-1}}^2 . \qquad (28)
$$

Denote $\boldsymbol{V} \triangleq \frac{\lambda}{\kappa_\mu}\mathbf{I}$. Note that the sequence of observation noises $\{\epsilon_s\}$ is 1-sub-Gaussian.

Next, we can apply Theorem 1 from (Abbasi-Yadkori et al., 2011), to obtain

$$
\left\| \sum_{\substack{s\in[t-1]:\\i_s=i}} \epsilon_s\tilde{\phi}_s \right\|_{\boldsymbol{V}_{i,t-1}^{-1}}^2 \leq 2\log\left( \frac{\det(\boldsymbol{V}_{i,t-1})^{1/2}}{\delta\det(\boldsymbol{V})^{1/2}} \right), \qquad (29)
$$

which holds with probability of at least $1 - \delta$.

Next, based on our assumption that $\left\|\tilde{\phi}_s\right\|_2 \leq 2$, according to Lemma 10 from (Abbasi-Yadkori et al., 2011), we have that

$$
\det(\boldsymbol{V}_{i,t-1}) \leq \left(\lambda/\kappa_\mu + 4T_{i,t}/d\right)^d , \qquad (30)
$$

where $T_{i,t}$ denotes the number of rounds of serving user $i$ in the first $t$ rounds. Therefore,

$$\sqrt{\frac{\det \boldsymbol{V}_{i,t-1}}{\det(V)}} \leq \sqrt{\frac{(\lambda/\kappa_\mu + 4T_{i,t}/d)^d}{(\lambda/\kappa_\mu)^d}} = (1 + 4T_{i,t}\kappa_\mu/(d\lambda))^{\frac{d}{2}} \tag{31}$$

This gives us

$$\left\|\sum_{\substack{s \in [t-1]: \\ i_s = i}} \epsilon_s \widetilde{\phi}_s\right\|_{\boldsymbol{V}_{i,t-1}^{-1}}^2 \leq 2\log\left(\frac{\det(\boldsymbol{V}_{i,t-1})^{1/2}}{\delta \det(V)^{1/2}}\right) \leq 2\log(1/\delta) + d\log\left(1 + 4T_{i,t}\kappa_\mu/(d\lambda)\right) \tag{32}$$

Then, with the above reasoning, we have that with probability at least $1 - \delta$ for some $\delta \in (0,1)$, at any $t \in [T]$:

$$\left\|\hat{\boldsymbol{\theta}}_{i,t} - \boldsymbol{\theta}^{j(i)}\right\|_{\boldsymbol{V}_{i,t-1}} \leq \frac{\sqrt{\lambda\kappa_\mu} + \sqrt{2\log(1/\delta) + d\log(1 + 4T_{i,t}\kappa_\mu/d\lambda)}}{\kappa_\mu}, \tag{33}$$

Taking a union bound over $u$ users, we have that with probability at least $1 - \delta$ for some $\delta \in (0,1)$, at any $t \in [T]$:

$$\left\|\hat{\boldsymbol{\theta}}_{i,t} - \boldsymbol{\theta}^{j(i)}\right\|_{\boldsymbol{V}_{i,t-1}} \leq \frac{\sqrt{\lambda\kappa_\mu} + \sqrt{2\log(u/\delta) + d\log(1 + 4T_{i,t}\kappa_\mu/d\lambda)}}{\kappa_\mu}, \forall i \in \mathcal{U}. \tag{34}$$

Then we have that with probability at least $1 - \delta$ for all $t \in [T]$ and all $i \in \mathcal{U}$

$$\begin{aligned}\left\|\hat{\boldsymbol{\theta}}_{i,t} - \boldsymbol{\theta}^{j(i)}\right\| &\leq \frac{\left\|\hat{\boldsymbol{\theta}}_{i,t} - \boldsymbol{\theta}^{j(i)}\right\|_{\boldsymbol{V}_{i,t-1}}}{\sqrt{\lambda_{\min(\boldsymbol{V}_{i,t-1})}}} \\ &\leq \frac{\sqrt{\lambda\kappa_\mu} + \sqrt{2\log(u/\delta) + d\log(1 + 4T_{i,t}\kappa_\mu/d\lambda)}}{\kappa_\mu\sqrt{\lambda_{\min(\boldsymbol{V}_{i,t-1})}}}.\end{aligned} \tag{35}$$

$\square$

Then, we prove the following lemma, which gives a sufficient time $T_0$ for the COLDB algorithm to cluster all the users correctly with high probability.

**Lemma B.2.** *With the carefully designed edge deletion rule, after*

$$T_0 \triangleq 16u\log(\frac{u}{\delta}) + 4u\max\{\frac{128d}{\kappa_\mu^2\tilde{\lambda}_x\gamma^2}\log(\frac{u}{\delta}), \frac{16}{\tilde{\lambda}_x^2}\log(\frac{8ud}{\tilde{\lambda}_x^2\delta})\}$$

$$= O\left(u\left(\frac{d}{\kappa_\mu^2\tilde{\lambda}_x\gamma^2} + \frac{1}{\tilde{\lambda}_x^2}\right)\log\frac{1}{\delta}\right)$$

*rounds, with probability at least $1 - 3\delta$ for some $\delta \in (0, \frac{1}{3})$, COLDB can cluster all the users correctly.*

*Proof.* Then, with the item regularity assumption stated in Assumption 2.4, Lemma J.1 in (Wang et al., 2024a), together with Lemma 7 in (Li & Zhang, 2018), and applying a union bound, with probability at least $1 - \delta$, for all $i \in \mathcal{U}$, at any $t$ such that $T_{i,t} \geq \frac{16}{\tilde{\lambda}_x^2}\log(\frac{8ud}{\tilde{\lambda}_x^2\delta})$, we have:

$$\lambda_{\min}(\boldsymbol{V}_{i,t}) \geq 2\tilde{\lambda}_x T_{i,t}. \tag{36}$$

Then, together with Lemma B.1, we have: if $T_{i,t} \geq \frac{16}{\tilde{\lambda}_x^2}\log(\frac{8ud}{\tilde{\lambda}_x^2\delta})$, then with probability $\geq 1 - 2\delta$, we have:

$$\left\|\hat{\boldsymbol{\theta}}_{i,t} - \boldsymbol{\theta}^{j(i)}\right\| \leq \frac{\sqrt{\lambda\kappa_\mu} + \sqrt{2\log(u/\delta) + d\log(1 + 4T_{i,t}\kappa_\mu/d\lambda)}}{\kappa_\mu\sqrt{\lambda_{\min(\boldsymbol{V}_{i,t-1})}}}$$

$$\leq \frac{\sqrt{\lambda \kappa_\mu} + \sqrt{2\log(u/\delta) + d\log(1 + 4T_{i,t}\kappa_\mu/d\lambda)}}{\kappa_\mu\sqrt{2\tilde{\lambda}_x T_{i,t}}} \ .$$

Now, let

$$\frac{\sqrt{\lambda \kappa_\mu} + \sqrt{2\log(u/\delta) + d\log(1 + 4T_{i,t}\kappa_\mu/d\lambda)}}{\kappa_\mu\sqrt{2\tilde{\lambda}_x T_{i,t}}} < \frac{\gamma}{4}, \tag{37}$$

Let $\lambda \kappa_\mu \leq 2\log(u/\delta) + d\log(1 + 4T_{i,t}\kappa_\mu/d\lambda)$, which typically holds ($\kappa_\mu$ is typically very small), we can get

$$\frac{2\log(u/\delta) + d\log(1 + 4T_{i,t}\kappa_\mu/d\lambda)}{2\kappa_\mu^2 \tilde{\lambda}_x T_{i,t}} < \frac{\gamma^2}{64}, \tag{38}$$

and a sufficient condition for it to hold is

$$\frac{2\log(u/\delta)}{2\kappa_\mu^2 \tilde{\lambda}_x T_{i,t}} < \frac{\gamma^2}{128} \tag{39}$$

and

$$\frac{d\log(1 + 4T_{i,t}\kappa_\mu/d\lambda)}{2\kappa_\mu^2 \tilde{\lambda}_x T_{i,t}} < \frac{\gamma^2}{128} \ . \tag{40}$$

Solving Eq.(39), we can get

$$T_{i,t} \geq \frac{128\log(u/\delta)}{\kappa_\mu^2 \tilde{\lambda}_x \gamma^2} \ . \tag{41}$$

Following Lemma 9 in (Li & Zhang, 2018), we can get the following sufficient condition for Eq.(40):

$$T_{i,t} \geq \frac{128d}{\kappa_\mu^2 \tilde{\lambda}_x \gamma^2} \log\Big(\frac{512}{\lambda \kappa_\mu \tilde{\lambda}_x \gamma^2}\Big) \ . \tag{42}$$

Let $u/\delta \geq 512/\lambda \kappa_\mu \tilde{\lambda}_x \gamma^2$, which is typically held. Then, combining all together, we have that if

$$T_{i,t} \geq \max\Big\{\frac{128d}{\kappa_\mu^2 \tilde{\lambda}_x \gamma^2} \log\Big(\frac{u}{\delta}\Big), \frac{16}{\tilde{\lambda}_x^2} \log\Big(\frac{8ud}{\tilde{\lambda}_x^2 \delta}\Big)\Big\}, \forall i \in \mathcal{U}, \tag{43}$$

then with probability at least $1 - 2\delta$, we have

$$\Big\|\hat{\boldsymbol{\theta}}_{i,t} - \boldsymbol{\theta}^{j(i)}\Big\| < \gamma/4, \forall i \in \mathcal{U}. \tag{44}$$

By Lemma 8 in (Li & Zhang, 2018), and Assumption 2.3 of user arrival uniformness, we have that for all

$$T_0 \triangleq 16u \log\Big(\frac{u}{\delta}\Big) + 4u \max\Big\{\frac{128d}{\kappa_\mu^2 \tilde{\lambda}_x \gamma^2} \log\Big(\frac{u}{\delta}\Big), \frac{16}{\tilde{\lambda}_x^2} \log\Big(\frac{8ud}{\tilde{\lambda}_x^2 \delta}\Big)\Big\}$$

$$= O\Big(u\Big(\frac{d}{\kappa_\mu^2 \tilde{\lambda}_x \gamma^2} + \frac{1}{\tilde{\lambda}_x^2}\Big) \log\frac{1}{\delta}\Big),$$

the condition in Eq.(43) is satisfied with probability at least $1 - \delta$.

Therefore we have that for all $t \geq T_0$, with probability $\geq 1 - 3\delta$:

$$\Big\|\hat{\boldsymbol{\theta}}_{i,t} - \boldsymbol{\theta}^{j(i)}\Big\|_2 < \frac{\gamma}{4}, \forall i \in \mathcal{U}. \tag{45}$$

Finally, we only need to show that with $\Big\|\hat{\boldsymbol{\theta}}_{i,t} - \boldsymbol{\theta}^{j(i)}\Big\|_2 < \frac{\gamma}{4}, \forall i \in \mathcal{U}$, the algorithm can cluster all the users correctly. First, when the edge $(i, l)$ is deleted, user $i$ and user $j$ must belong to different *ground-truth clusters*, i.e., $\|\boldsymbol{\theta}_i - \boldsymbol{\theta}_l\|_2 > 0$. This is because by the deletion rule of the algorithm, the concentration bound, and triangle inequality

$$\|\boldsymbol{\theta}_i - \boldsymbol{\theta}_l\|_2 = \Big\|\boldsymbol{\theta}^{j(i)} - \boldsymbol{\theta}^{j(l)}\Big\|_2$$

$$\geq \left\| \hat{\boldsymbol{\theta}}_{i,t} - \hat{\boldsymbol{\theta}}_{l,t} \right\|_2 - \left\| \boldsymbol{\theta}^{j(l)} - \hat{\boldsymbol{\theta}}_{l,t} \right\|_2 - \left\| \boldsymbol{\theta}^{j(i)} - \hat{\boldsymbol{\theta}}_{i,t} \right\|_2$$

$$\geq \left\| \hat{\boldsymbol{\theta}}_{i,t} - \hat{\boldsymbol{\theta}}_{l,t} \right\|_2 - f(T_{i,t}) - f(T_{l,t}) > 0 \,. \tag{46}$$

Second, we can show that if $\|\boldsymbol{\theta}_i - \boldsymbol{\theta}_l\| > \gamma$, meaning that user $i$ and user $l$ are not in the same *ground-truth cluster*, COLDB will delete the edge $(i,l)$ after $T_0$. This is because

$$\left\| \hat{\boldsymbol{\theta}}_{i,t} - \hat{\boldsymbol{\theta}}_{l,t} \right\| \geq \|\boldsymbol{\theta}_i - \boldsymbol{\theta}_l\| - \left\| \hat{\boldsymbol{\theta}}_{i,t} - \boldsymbol{\theta}^{j(i)} \right\|_2 - \left\| \hat{\boldsymbol{\theta}}_{l,t} - \boldsymbol{\theta}^{j(l)} \right\|_2$$

$$> \gamma - \frac{\gamma}{4} - \frac{\gamma}{4}$$

$$= \frac{\gamma}{2} > f(T_{i,t}) + f(T_{l,t}) \,, \tag{47}$$

which will trigger the edge deletion rule to delete edge $(i,l)$. Combining all the reasoning above, we can finish the proof. $\quad\square$

Then, we prove the following lemmas for the cluster-based statistics.

**Lemma B.3.** *With probability at least $1 - 4\delta$ for some $\delta \in (0, 1/4)$, at any $t \geq T_0$:*

$$\left\| \overline{\boldsymbol{\theta}}_t - \boldsymbol{\theta}_{i_t} \right\|_{\boldsymbol{V}_{t-1}} \leq \frac{\sqrt{\lambda \kappa_\mu} + \sqrt{2 \log(u/\delta) + d \log(1 + 4T\kappa_\mu/d\lambda)}}{\kappa_\mu} \,. \tag{48}$$

*Proof.* First, by Lemma B.2, we have that with probability at least $1 - 3\delta$, all the users are clustered correctly, i.e., $\overline{C}_t = C_{j(i_t)}, \forall t \geq T_0$. Recall that $f_i(\boldsymbol{x}) = \boldsymbol{\theta}_i^\top \phi(\boldsymbol{x})$. In iteration $s$, define $\widetilde{\phi}_s = \phi(\boldsymbol{x}_{s,1}) - \phi(\boldsymbol{x}_{s,2})$. And we define $\widetilde{f}_{i,s} = f_i(\boldsymbol{x}_{s,1}) - f_i(\boldsymbol{x}_{s,2}) = \boldsymbol{\theta}_i^\top \widetilde{\phi}_s$.

For any $\boldsymbol{\theta}_{f'} \in \mathbb{R}^d$, define

$$G_t(\boldsymbol{\theta}_{f'}) = \sum_{\substack{s \in [t-1]:\\ i_s \in \overline{C}_t}} \left( \mu(\boldsymbol{\theta}_{f'}^\top \widetilde{\phi}_s) - \mu(\boldsymbol{\theta}_{i_t}^\top \widetilde{\phi}_s) \right) \widetilde{\phi}_s + \lambda \boldsymbol{\theta}_{f'} \,.$$

For $\lambda' \in (0, 1)$, setting $\boldsymbol{\theta}_{\bar{f}} = \lambda' \boldsymbol{\theta}_{f_1'} + (1 - \lambda') \boldsymbol{\theta}_{f_2'}$. and using the mean-value theorem, we get:

$$G_t(\boldsymbol{\theta}_{f_1'}) - G_t(\boldsymbol{\theta}_{f_2'}) = \left[ \sum_{\substack{s \in [t-1]:\\ i_s \in \overline{C}_t}} \nabla \mu(\boldsymbol{\theta}_{\bar{f}}^\top \widetilde{\phi}_s) \widetilde{\phi}_s \widetilde{\phi}_s^\top + \lambda \mathbf{I} \right] (\boldsymbol{\theta}_{f_1'} - \boldsymbol{\theta}_{f_2'}) \tag{49}$$

Define $\boldsymbol{M}_{t-1} = \left[ \sum_{\substack{s \in [t-1]:\\ i_s \in \overline{C}_t}} \nabla \mu(\boldsymbol{\theta}_{\bar{f}}^\top \widetilde{\phi}_s) \widetilde{\phi}_s \widetilde{\phi}_s^\top + \lambda \mathbf{I} \right]$, and recall that $\boldsymbol{V}_{t-1} = \sum_{\substack{s \in [t-1]:\\ i_s \in \overline{C}_t}} \widetilde{\phi}_s \widetilde{\phi}_s^\top + \frac{\lambda}{\kappa_\mu} \mathbf{I}$. Then we have that $\boldsymbol{M}_{t-1} \succeq \kappa_\mu \boldsymbol{V}_{t-1}$ and that $\boldsymbol{V}_{t-1}^{-1} \succeq \kappa_\mu \boldsymbol{M}_{t-1}^{-1}$. Then we have

$$
\begin{aligned}
\left\| G_t(\overline{\boldsymbol{\theta}}_t) - \lambda \boldsymbol{\theta}_{i_t} \right\|_{\boldsymbol{V}_{t-1}^{-1}}^2 &= \left\| G_t(\boldsymbol{\theta}_{i_t}) - G_t(\overline{\boldsymbol{\theta}}_t) \right\|_{\boldsymbol{V}_{t-1}^{-1}}^2 = \left\| \boldsymbol{M}_{t-1}(\boldsymbol{\theta}_{i_t} - \overline{\boldsymbol{\theta}}_t) \right\|_{\boldsymbol{V}_{t-1}^{-1}}^2 && (G_t(\boldsymbol{\theta}_{i_t}) = \lambda \boldsymbol{\theta}_{i_t} \text{ by definition})\\
&= (\boldsymbol{\theta}_{i_t} - \overline{\boldsymbol{\theta}}_t)^\top \boldsymbol{M}_{t-1} \boldsymbol{V}_{t-1}^{-1} \boldsymbol{M}_{t-1} (\boldsymbol{\theta}_{i_t} - \overline{\boldsymbol{\theta}}_t)\\
&\geq (\boldsymbol{\theta}_{i_t} - \overline{\boldsymbol{\theta}}_t)^\top \boldsymbol{M}_{t-1} \kappa_\mu \boldsymbol{M}_{t-1}^{-1} \boldsymbol{M}_{t-1} (\boldsymbol{\theta}_{i_t} - \overline{\boldsymbol{\theta}}_t)\\
&= \kappa_\mu (\boldsymbol{\theta}_{i_t} - \overline{\boldsymbol{\theta}}_t)^\top \boldsymbol{M}_{t-1} (\boldsymbol{\theta}_{i_t} - \overline{\boldsymbol{\theta}}_t)\\
&\geq \kappa_\mu (\boldsymbol{\theta}_{i_t} - \overline{\boldsymbol{\theta}}_t)^\top \kappa_\mu \boldsymbol{V}_{t-1} (\boldsymbol{\theta}_{i_t} - \overline{\boldsymbol{\theta}}_t)\\
&= \kappa_\mu^2 (\boldsymbol{\theta}_{i_t} - \overline{\boldsymbol{\theta}}_t)^\top \boldsymbol{V}_{t-1} (\boldsymbol{\theta}_{i_t} - \overline{\boldsymbol{\theta}}_t)\\
&= \kappa_\mu^2 \left\| \boldsymbol{\theta}_{i_t} - \overline{\boldsymbol{\theta}}_t \right\|_{\boldsymbol{V}_{t-1}}^2 && (\text{as } \|\boldsymbol{x}\|_{\boldsymbol{A}}^2 = \boldsymbol{x}^\top \boldsymbol{A} \boldsymbol{x})
\end{aligned}
$$

The first inequality is because $\boldsymbol{V}_{t-1}^{-1} \succeq \kappa_\mu \boldsymbol{M}_{t-1}^{-1}$, and the second inequality follows from $\boldsymbol{M}_{t-1} \succeq \kappa_\mu \boldsymbol{V}_{t-1}$.

Note that $\frac{\kappa_\mu}{\lambda}\mathbf{I} \succeq \mathbf{V}_{t-1}$, which allows us to show that

$$\|\lambda\boldsymbol{\theta}_{i_t}\|_{\mathbf{V}_{t-1}^{-1}} = \lambda\sqrt{\boldsymbol{\theta}_{i_t}^\top \mathbf{V}_{t-1}^{-1}\boldsymbol{\theta}_{i_t}} \leq \lambda\sqrt{\boldsymbol{\theta}_{i_t}^\top \frac{\kappa_\mu}{\lambda}\boldsymbol{\theta}_{i_t}} \leq \sqrt{\lambda\kappa_\mu}\|\boldsymbol{\theta}_{i_t}\|_2 \leq \sqrt{\lambda\kappa_\mu}. \tag{50}$$

Using the two equations above, we have that

$$\left\|\boldsymbol{\theta}_{i_t} - \overline{\boldsymbol{\theta}}_t\right\|_{\mathbf{V}_{t-1}} \leq \frac{1}{\kappa_\mu}\left\|G_t(\overline{\boldsymbol{\theta}}_t) - \lambda\boldsymbol{\theta}_{i_t}\right\|_{\mathbf{V}_{t-1}^{-1}} \leq \frac{1}{\kappa_\mu}\left\|G_t(\overline{\boldsymbol{\theta}}_t)\right\|_{\mathbf{V}_{t-1}^{-1}} + \frac{1}{\kappa_\mu}\left\|\lambda\boldsymbol{\theta}_{i_t}\right\|_{\mathbf{V}_{t-1}^{-1}}$$
$$\leq \frac{1}{\kappa_\mu}\left\|G_t(\overline{\boldsymbol{\theta}}_t)\right\|_{\mathbf{V}_{t-1}^{-1}} + \sqrt{\frac{\lambda}{\kappa_\mu}} \tag{51}$$

Then, let $\overline{f}_{t,s} = \overline{\boldsymbol{\theta}}_t^\top \widetilde{\phi}_s$, we have:

$$\frac{1}{\kappa_\mu^2}\left\|G_t(\overline{\boldsymbol{\theta}}_t)\right\|_{\mathbf{V}_{t-1}^{-1}}^2 \leq \frac{1}{\kappa_\mu^2}\left\|\sum_{\substack{s\in[t-1]:\\ i_s\in\overline{C}_t}} (\mu(\overline{\boldsymbol{\theta}}_t^\top\widetilde{\phi}_s) - \mu(\boldsymbol{\theta}_{i_t}^\top\widetilde{\phi}_s))\widetilde{\phi}_s + \lambda\overline{\boldsymbol{\theta}}_t\right\|_{\mathbf{V}_{t-1}^{-1}}^2 \qquad \text{(by definition of } G_t(\overline{\boldsymbol{\theta}}_t)\text{)}$$

$$= \frac{1}{\kappa_\mu^2}\left\|\sum_{\substack{s\in[t-1]:\\ i_s\in\overline{C}_t}} (\mu(\overline{f}_{t,s}) - \mu(\widetilde{f}_{i_t,s}))\widetilde{\phi}_s + \lambda\overline{\boldsymbol{\theta}}_t\right\|_{\mathbf{V}_{t-1}^{-1}}^2 \qquad \left(\text{see definitions of } \overline{f}_{t,s} \text{ and } \widetilde{f}_{i,s}\right)$$

$$= \frac{1}{\kappa_\mu^2}\left\|\sum_{\substack{s\in[t-1]:\\ i_s\in\overline{C}_t}} (\mu(\overline{f}_{t,s}) - (y_s - \epsilon_s))\widetilde{\phi}_s + \lambda\overline{\boldsymbol{\theta}}_t\right\|_{\mathbf{V}_{t-1}^{-1}}^2$$

$$\left(y_s = \mu(\widetilde{f}_{i_t,s}) + \epsilon_s \text{ if } i_s = i_t, \text{ and } i_s = i_t, \forall i_s \in \overline{C}_t, \forall t \geq T_0\right)$$

$$= \frac{1}{\kappa_\mu^2}\left\|\sum_{\substack{s\in[t-1]:\\ i_s\in\overline{C}_t}} \left(\mu(\overline{f}_{t,s}) - y_s\right)\widetilde{\phi}_s + \sum_{\substack{s\in[t-1]:\\ i_s\in\overline{C}_t}} \epsilon_s\widetilde{\phi}_s + \lambda\overline{\boldsymbol{\theta}}_t\right\|_{\mathbf{V}_{t-1}^{-1}}^2$$

$$\leq \frac{1}{\kappa_\mu^2}\left\|\sum_{\substack{s\in[t-1]:\\ i_s\in\overline{C}_t}} \epsilon_s\widetilde{\phi}_s\right\|_{\mathbf{V}_{t-1}^{-1}}^2 .$$

The last step holds due to the following reasoning. Recall that $\overline{\boldsymbol{\theta}}_t$ is computed using MLE by solving the following equation:

$$\overline{\boldsymbol{\theta}}_t = \arg\min_{\boldsymbol{\theta}} - \sum_{\substack{s\in[t-1]\\ i_s\in\overline{C}_t}} \left(y_s \log\mu\left(\boldsymbol{\theta}^\top[\phi(\boldsymbol{x}_{s,1}) - \phi(\boldsymbol{x}_{s,2})]\right) + (1-y_s)\log\mu\left(\boldsymbol{\theta}^\top[\phi(\boldsymbol{x}_{s,2}) - \phi(\boldsymbol{x}_{s,1})]\right)\right) + \frac{1}{2}\lambda\|\boldsymbol{\theta}\|_2^2. \tag{52}$$

Setting its gradient to 0, the following is satisfied:

$$\sum_{\substack{s\in[t-1]:\\ i_s\in\overline{C}_t}} \left(\mu\left(\overline{\boldsymbol{\theta}}_t^\top\widetilde{\phi}_s\right) - y_s\right)\widetilde{\phi}_s + \lambda\overline{\boldsymbol{\theta}}_t = 0, \tag{53}$$

which is used in the last step.

Now we have

$$\left\|\boldsymbol{\theta}_{i_t} - \overline{\boldsymbol{\theta}}_t\right\|_{\mathbf{V}_{t-1}} \leq \frac{1}{\kappa_\mu}\left\|\sum_{\substack{s\in[t-1]:\\ i_s\in\overline{C}_t}} \epsilon_s\widetilde{\phi}_s\right\|_{\mathbf{V}_{t-1}^{-1}} + \sqrt{\frac{\lambda}{\kappa_\mu}}. \tag{54}$$

Denote $V \triangleq \frac{\lambda}{\kappa_\mu} I$. Note that the sequence of observation noises $\{\epsilon_s\}$ is 1-sub-Gaussian.

Next, we can apply Theorem 1 from (Abbasi-Yadkori et al., 2011), to obtain

$$\left\| \sum_{\substack{s \in [t-1]: \\ i_s \in \overline{C}_t}} \epsilon_s \widetilde{\phi}_s \right\|_{V_{t-1}^{-1}}^2 \leq 2 \log \left( \frac{\det(V_{t-1})^{1/2}}{\delta \det(V)^{1/2}} \right), \tag{55}$$

which holds with probability of at least $1 - \delta$.

Next, based on our assumption that $\left\| \widetilde{\phi}_s \right\|_2 \leq 2$, according to Lemma 10 from (Abbasi-Yadkori et al., 2011), we have that

$$\det(V_{t-1}) \leq (\lambda/\kappa_\mu + 4T/d)^d. \tag{56}$$

Therefore,

$$\sqrt{\frac{\det V_{t-1}}{\det(V)}} \leq \sqrt{\frac{(\lambda/\kappa_\mu + 4T/d)^d}{(\lambda/\kappa_\mu)^d}} = (1 + 4T\kappa_\mu/(d\lambda))^{\frac{d}{2}} \tag{57}$$

This gives us

$$\left\| \sum_{\substack{s \in [t-1]: \\ i_s \in \overline{C}_t}} \epsilon_s \widetilde{\phi}_s \right\|_{V_{t-1}^{-1}}^2 \leq 2 \log \left( \frac{\det(V_{t-1})^{1/2}}{\delta \det(V)^{1/2}} \right) \leq 2 \log(1/\delta) + d \log \left( 1 + 4T\kappa_\mu/(d\lambda) \right) \tag{58}$$

Combining all together, we have with probability at least $1 - 4\delta$ for some $\delta \in (0, 1/4)$, at any $t \geq T_0$:

$$\left\| \overline{\theta}_t - \theta_{i_t} \right\|_{V_{t-1}} \leq \frac{\sqrt{\lambda \kappa_\mu} + \sqrt{2 \log(u/\delta) + d \log(1 + 4T\kappa_\mu/d\lambda)}}{\kappa_\mu}. \tag{59}$$

$\square$

Then, we prove the following lemma with the help of Lemma B.3.

**Lemma B.4.** *For any iteration $t \geq T_0$, for all $x, x' \in \mathcal{X}_t$, with probability of at least $1 - 4\delta$, we have*

$$\left| (f_{i_t}(x) - f_{i_t}(x')) - \overline{\theta}_t^\top (\phi(x) - \phi(x')) \right| \leq \frac{\beta_T}{\kappa_\mu} \left\| \phi(x) - \phi(x') \right\|_{V_{t-1}^{-1}},$$

*where $\beta_T = \sqrt{\lambda \kappa_\mu} + \sqrt{2 \log(u/\delta) + d \log(1 + 4T\kappa_\mu/d\lambda)}$.*

*Proof.*

$$\begin{aligned}
\left| (f_{i_t}(x) - f_{i_t}(x')) - \overline{\theta}_t^\top (\phi(x) - \phi(x')) \right| &= \left| \theta_{i_t}^\top [(\phi(x) - \phi(x')] - \overline{\theta}_t^\top [\phi(x) - \phi(x')] \right| \\
&= \left| (\theta_{i_t} - \overline{\theta}_t)^\top [\phi(x) - \phi(x')] \right| \\
&\leq \left\| \theta_{i_t} - \overline{\theta}_t \right\|_{V_{t-1}} \left\| \phi(x) - \phi(x') \right\|_{V_{t-1}^{-1}} \\
&\leq \frac{\beta_T}{\kappa_\mu} \left\| \phi(x) - \phi(x') \right\|_{V_{t-1}^{-1}},
\end{aligned} \tag{60}$$

in which the last inequality follows from Lemma B.3. $\square$

We also prove the following lemma to upper bound the summation of squared norms which will be used in proving the final regret bound.

**Lemma B.5.** *With probability at least $1 - 4\delta$, we have*

$$\sum_{t=T_0}^{T} \mathbb{I}\{i_t \in C_j\} \|\phi(\boldsymbol{x}_{t,1}) - \phi(\boldsymbol{x}_{t,2})\|_{\boldsymbol{V}_{t-1}^{-1}}^2 \le 2d \log \left(1 + 4T\kappa_\mu/(d\lambda)\right), \forall j \in [m],$$

*where $\mathbb{I}$ denotes the indicator function.*

*Proof.* We denote $\widetilde{\phi}_t = \phi(\boldsymbol{x}_{t,1}) - \phi(\boldsymbol{x}_{t,2})$. Recall that we have assumed that $\|\phi(\boldsymbol{x}_{t,1}) - \phi(\boldsymbol{x}_{t,2})\|_2 \le 2$. It is easy to verify that $\boldsymbol{V}_{t-1} \succeq \frac{\lambda}{\kappa_\mu} I$ and hence $\boldsymbol{V}_{t-1}^{-1} \preceq \frac{\kappa_\mu}{\lambda} I$. Therefore, we have that $\left\|\widetilde{\phi}_t\right\|_{\boldsymbol{V}_{t-1}^{-1}}^2 \le \frac{\kappa_\mu}{\lambda} \left\|\widetilde{\phi}_t\right\|_2^2 \le \frac{4\kappa_\mu}{\lambda}$. We choose $\lambda$ such that $\frac{4\kappa_\mu}{\lambda} \le 1$, which ensures that $\left\|\widetilde{\phi}_t\right\|_{\boldsymbol{V}_{t-1}^{-1}}^2 \le 1$. Our proof here mostly follows from Lemma 11 of (Abbasi-Yadkori et al., 2011) and Lemma J.2 of (Wang et al., 2024a). To begin with, note that $x \le 2\log(1 + x)$ for $x \in [0, 1]$. Denote $\boldsymbol{V}_{t,j} = \sum_{\substack{s \in [t-1]: \\ i_s \in C_j}} \widetilde{\phi}_s \widetilde{\phi}_s^\top + \frac{\lambda}{\kappa_\mu} \mathbf{I}$. Then we have that

$$\begin{aligned}
\sum_{t=T_0}^{T} \mathbb{I}\{i_t \in C_j\} \left\|\widetilde{\phi}_t\right\|_{\boldsymbol{V}_{t-1}^{-1}}^2 &\le \sum_{t=T_0}^{T} 2\log \left(1 + \mathbb{I}\{i_t \in C_j\} \left\|\widetilde{\phi}_t\right\|_{\boldsymbol{V}_{t-1}^{-1}}^2\right) \\
&= 2\left(\log \det V_{T,j} - \log \det V\right) \\
&= 2\log \frac{\det V_{T,j}}{\det V} \\
&\le 2\log \left((1 + 4T\kappa_\mu/(d\lambda))^d\right) \\
&= 2d \log \left(1 + 4T\kappa_\mu/(d\lambda)\right).
\end{aligned} \tag{61}$$

The second inequality follows the same reasoning as equation (57). This completes the proof. $\square$

Now we are ready to prove Theorem 4.1. First, we have

$$R_T = \sum_{t=1}^{T} r_t \le T_0 + \sum_{t=T_0}^{T} r_t, \tag{62}$$

where we use that the reward at each round is bounded by 1.

Then, we only need to upper bound the regret after $T_0$. By Lemma B.2, we know that with probability at least $1 - 4\delta$, the algorithm can cluster all the users correctly, $\overline{C}_t = C_{j(i_t)}$, and the statements of all the above lemmas hold. We have that for

any $t \geq T_0$:

$$r_t = f_{i_t}(\boldsymbol{x}_t^*) - f_{i_t}(\boldsymbol{x}_{t,1}) + f_{i_t}(\boldsymbol{x}_t^*) - f_{i_t}(x_{t,2})$$

$$\overset{(a)}{\leq} \overline{\boldsymbol{\theta}}_t^\top (\phi(\boldsymbol{x}_t^*) - \phi(\boldsymbol{x}_{t,1})) + \frac{\beta_T}{\kappa_\mu} \|\phi(\boldsymbol{x}_t^*) - \phi(\boldsymbol{x}_{t,1})\|_{\boldsymbol{V}_{t-1}^{-1}} + \overline{\boldsymbol{\theta}}_t^\top (\phi(\boldsymbol{x}_t^*) - \phi(\boldsymbol{x}_{t,2})) + \frac{\beta_T}{\kappa_\mu} \|\phi(\boldsymbol{x}_t^*) - \phi(\boldsymbol{x}_{t,2})\|_{\boldsymbol{V}_{t-1}^{-1}}$$

$$= \overline{\boldsymbol{\theta}}_t^\top (\phi(\boldsymbol{x}_t^*) - \phi(\boldsymbol{x}_{t,1})) + \frac{\beta_T}{\kappa_\mu} \|\phi(\boldsymbol{x}_t^*) - \phi(\boldsymbol{x}_{t,1})\|_{\boldsymbol{V}_{t-1}^{-1}} +$$

$$\qquad \overline{\boldsymbol{\theta}}_t^\top (\phi(\boldsymbol{x}_t^*) - \phi(\boldsymbol{x}_{t,1})) + \overline{\boldsymbol{\theta}}_t^\top (\phi(\boldsymbol{x}_{t,1}) - \phi(\boldsymbol{x}_{t,2})) + \frac{\beta_T}{\kappa_\mu} \|\phi(\boldsymbol{x}_t^*) - \phi(\boldsymbol{x}_{t,1}) + \phi(\boldsymbol{x}_{t,1}) - \phi(\boldsymbol{x}_{t,2})\|_{\boldsymbol{V}_{t-1}^{-1}}$$

$$\overset{(b)}{\leq} 2\overline{\boldsymbol{\theta}}_t^\top (\phi(x^*) - \phi(\boldsymbol{x}_{t,1})) + 2\frac{\beta_T}{\kappa_\mu} \|\phi(x^*) - \phi(\boldsymbol{x}_{t,1})\|_{\boldsymbol{V}_{t-1}^{-1}} +$$

$$\qquad \overline{\boldsymbol{\theta}}_t^\top (\phi(\boldsymbol{x}_{t,1}) - \phi(\boldsymbol{x}_{t,2})) + \frac{\beta_T}{\kappa_\mu} \|\phi(\boldsymbol{x}_{t,1}) - \phi(\boldsymbol{x}_{t,2})\|_{\boldsymbol{V}_{t-1}^{-1}}$$

$$\overset{(c)}{\leq} 2\overline{\boldsymbol{\theta}}_t^\top (\phi(\boldsymbol{x}_{t,2}) - \phi(\boldsymbol{x}_{t,1})) + 2\frac{\beta_T}{\kappa_\mu} \|\phi(\boldsymbol{x}_{t,2}) - \phi(\boldsymbol{x}_{t,1})\|_{\boldsymbol{V}_{t-1}^{-1}} +$$

$$\qquad \overline{\boldsymbol{\theta}}_t^\top (\phi(\boldsymbol{x}_{t,1}) - \phi(\boldsymbol{x}_{t,2})) + \frac{\beta_T}{\kappa_\mu} \|\phi(\boldsymbol{x}_{t,1}) - \phi(\boldsymbol{x}_{t,2})\|_{\boldsymbol{V}_{t-1}^{-1}}$$

$$\leq \overline{\boldsymbol{\theta}}_t^\top (\phi(\boldsymbol{x}_{t,2}) - \phi(\boldsymbol{x}_{t,1})) + 3\frac{\beta_T}{\kappa_\mu} \|\phi(\boldsymbol{x}_{t,2}) - \phi(\boldsymbol{x}_{t,1})\|_{\boldsymbol{V}_{t-1}^{-1}}$$

$$\overset{(d)}{\leq} 3\frac{\beta_T}{\kappa_\mu} \|\phi(\boldsymbol{x}_{t,1}) - \phi(\boldsymbol{x}_{t,2})\|_{\boldsymbol{V}_{t-1}^{-1}}$$

(63)

Step $(a)$ follows from Lemma B.4. Step $(b)$ makes use of the triangle inequality. Step $(c)$ follows from the way in which we choose the second arm $\boldsymbol{x}_{t,2}$: $\boldsymbol{x}_{t,2} = \arg\max_{x \in \mathcal{X}_t} \overline{\boldsymbol{\theta}}_t^\top (\phi(x) - \phi(\boldsymbol{x}_{t,1})) + \frac{\beta_T}{\kappa_\mu} \|\phi(x) - \phi(\boldsymbol{x}_{t,1})\|_{\boldsymbol{V}_{t-1}^{-1}}$. Step $(d)$ results from the way in which we select the first arm: $\boldsymbol{x}_{t,1} = \arg\max_{x \in \mathcal{X}_t} \overline{\boldsymbol{\theta}}_t^\top \phi(x)$.

Then we have

$$\sum_{t=T_0}^{T} r_t \leq 3\frac{\beta_T}{\kappa_\mu} \sum_{t=T_0}^{T} \|\phi(\boldsymbol{x}_{t,1}) - \phi(\boldsymbol{x}_{t,2})\|_{\boldsymbol{V}_{t-1}^{-1}}$$

$$= 3\frac{\beta_T}{\kappa_\mu} \sum_{t=T_0}^{T} \sum_{j \in [m]} \mathbb{I}\{i_t \in C_j\} \|\phi(\boldsymbol{x}_{t,1}) - \phi(\boldsymbol{x}_{t,2})\|_{\boldsymbol{V}_{t-1}^{-1}}$$

$$\leq 3\frac{\beta_T}{\kappa_\mu} \sqrt{\sum_{t=T_0}^{T} \sum_{j \in [m]} \mathbb{I}\{i_t \in C_j\} \sum_{t=T_0}^{T} \sum_{j \in [m]} \mathbb{I}\{i_t \in C_j\} \|\phi(\boldsymbol{x}_{t,1}) - \phi(\boldsymbol{x}_{t,2})\|_{\boldsymbol{V}_{t-1}^{-1}}^2}$$

$$\leq 3\frac{\beta_T}{\kappa_\mu} \sqrt{T \cdot m \cdot 2d \log(1 + 4T\kappa_\mu/(d\lambda))},$$

(64)

where in the second inequality we use the Cauchy-Swarchz inequality, and in the last step we use $\sum_{t=T_0}^{T} \sum_{j \in [m]} \mathbb{I}\{i_t \in C_j\} \leq T$ and Lemma B.5.

Therefore, finally, we have with probability at least $1 - 4\delta$

$$R_T \leq T_0 + 3\frac{\beta_T}{\kappa_\mu} \sqrt{T \cdot m \cdot 2d \log(1 + 4T\kappa_\mu/(d\lambda))}$$

$$\leq O(u(\frac{d}{\kappa_\mu^2 \tilde{\lambda}_x \gamma^2} + \frac{1}{\tilde{\lambda}_x^2}) \log T + \frac{1}{\kappa_\mu} d\sqrt{mT})$$

$$= O(\frac{1}{\kappa_\mu} d\sqrt{mT}),$$

(65)

# C. Proof of Theorem 4.2

## C.1. Auxiliary Definitions and Explanations

**Denifition of the NTK matrix $\mathbf{H}_j$ for cluster $j$.** Recall that we use $T_j$ to denote the total number of iterations in which the users in cluster $j$ are served. For cluster $j$, let $\{x_{(i)}\}_{i=1}^{T_j K}$ be a set of all $T_j \times K$ possible arm feature vectors: $\{x_{t,a}\}_{1 \le t \le T_j, 1 \le a \le K}$, where $i = K(t-1)+a$. Firstly, we define $\mathbf{h}_t = [f^j(x_{(i)})]_{i=1,\dots,T_j K}^\top$, i.e., $\mathbf{h}_t$ is the $T_j K$-dimensional vector containing the reward function values of the arms corresponding to cluster $j$. Next, define

$$\widetilde{\mathbf{H}}_{p,q}^{(1)} = \mathbf{\Sigma}_{p,q}^{(1)} = \langle x_{(p)}, x_{(q)} \rangle, \quad \mathbf{A}_{p,q}^{(l)} = \begin{pmatrix} \mathbf{\Sigma}_{p,q}^{(l)} & \mathbf{\Sigma}_{p,q}^{(l)} \\ \mathbf{\Sigma}_{p,q}^{(l)} & \mathbf{\Sigma}_{q,q}^{(l)} \end{pmatrix},$$

$$\mathbf{\Sigma}_{p,q}^{(l+1)} = 2\mathbb{E}_{(u,v) \sim \mathcal{N}(0, \mathbf{A}_{p,q}^{(l)})}[\max\{u,0\}\max\{v,0\}],$$

$$\widetilde{\mathbf{H}}_{p,q}^{(l+1)} = 2\widetilde{\mathbf{H}}_{p,q}^{(l)} \mathbb{E}_{(u,v) \sim \mathcal{N}(0, \mathbf{A}_{p,q}^{(l)})}[\mathbb{1}(u \ge 0)\mathbb{1}(v \ge 0)] + \mathbf{\Sigma}_{p,q}^{(l+1)}.$$

With these definitions, the NTK matrix for cluster $j$ is then defined as $\mathbf{H}_j = (\widetilde{\mathbf{H}}^{(L)} + \mathbf{\Sigma}^{(L)})/2$.

**The Initial Parameters $\boldsymbol{\theta}_0$.** Next, we discuss how the initial parameters $\boldsymbol{\theta}_0$ are obtained. We adopt the same initialization method from Zhang et al. (2021); Zhou et al. (2020). Specifically, for each $l = 1, \dots, L-1$, let $\mathbf{W}_l = \begin{pmatrix} \mathbf{W} & \mathbf{0} \\ \mathbf{0} & \mathbf{W} \end{pmatrix}$ in which every entry of $\mathbf{W}$ is independently and randomly sampled from $\mathcal{N}(0, 4/m_{\mathrm{NN}})$, and choose $\mathbf{W}_L = (\mathbf{w}^\top, -\mathbf{w}^\top)$ in which every entry of $\mathbf{w}$ is independently and randomly sampled from $\mathcal{N}(0, 2/m_{\mathrm{NN}})$.

**Justifications for Assumption 2.5.** The last assumption in Assumption 2.5, together with the way we initialize $\theta_0$ as discussed above, ensures that the initial output of the NN is 0: $h(x; \boldsymbol{\theta}_0) = 0, \forall x \in \mathcal{X}$. The assumption of $x_j = x_{j+d/2}$ from Assumption 2.5 is a mild assumption which is commonly adopted by previous works on neural bandits (Zhou et al., 2020; Zhang et al., 2021). To ensure that this assumption holds, for any arm $x$, we can always firstly normalize it such that $||x|| = 1$, and then construct a new context $x' = (x^\top, x^\top)^\top/\sqrt{2}$ to satisfy this assumption (Zhou et al., 2020).

## C.2. Proof

To begin with, we first list the specific conditions we need for the width $m_{\mathrm{NN}}$ of the NN:

$$\begin{aligned}
& m_{\mathrm{NN}} \ge CT^4 K^4 L^6 \log(T^2 K^2 L/\delta)/\lambda_0^4, \\
& m_{\mathrm{NN}}(\log m)^{-3} \ge C\kappa_\mu^{-3} T^8 L^{21} \lambda^{-5}, \\
& m_{\mathrm{NN}}(\log m_{\mathrm{NN}})^{-3} \ge C\kappa_\mu^{-3} T^{14} L^{21} \lambda^{-11} L_\mu^6, \\
& m_{\mathrm{NN}}(\log m_{\mathrm{NN}})^{-3} \ge CT^{14} L^{18} \lambda^{-8},
\end{aligned} \tag{66}$$

for some absolute constant $C > 0$. To ease exposition, we express these conditions above as $m_{\mathrm{NN}} \ge \mathrm{poly}(T, L, K, 1/\kappa_\mu, L_\mu, 1/\lambda_0, 1/\lambda, \log(1/\delta))$.

In our proof here, we use the gradient of the NN at $\theta_0$ to derive the feature mapping for the arms, i.e., we let $\phi(\boldsymbol{x}) = g(\boldsymbol{x}; \boldsymbol{\theta}_0)/\sqrt{m_{\mathrm{NN}}}$. We use $\hat{\boldsymbol{\theta}}_{i,t}$ to denote the paramters of the NN after training in iteration $t$ (see Algorithm 2).

We use the following lemma to show that for every cluster $j \in \mathcal{C}$, its reward function $f^j$ can be expressed as a linear function with respect to the initial gradient $g(\boldsymbol{x}; \boldsymbol{\theta}_0)$.

**Lemma C.1** (Lemma B.3 of (Zhang et al., 2021)). *As long as the width $m$ of the NN is large enough:*

$$m_{NN} \ge C_0 T^4 K^4 L^6 \log(T^2 K^2 L/\delta)/\lambda_0^4,$$

*then for all clusters $j \in [m]$, with probability of at least $1 - \delta$, there exits a $\boldsymbol{\theta}_f^j$ such that*

$$f^j(\boldsymbol{x}) = \langle g(\boldsymbol{x}; \boldsymbol{\theta}_0), \boldsymbol{\theta}_f^j - \boldsymbol{\theta}_0 \rangle, \qquad \sqrt{m_{NN}} \left\| \boldsymbol{\theta}_f^j - \boldsymbol{\theta}_0 \right\|_2 \le \sqrt{2\mathbf{h}_j^\top \mathbf{H}_j^{-1} \mathbf{h}_j} \le B.$$

*for all $\boldsymbol{x} \in \mathcal{X}_t$, $t \in [T]$ with $i_t \in C_j$.*

Lemma C.1 is the formal statement of Lemma 2.6 from Sec. 2.2. Note that the constant $B$ is applicable to all $m$ clusters.

The following lemma converts our assumption about cluster separation (Assumption 2.7) into the difference between the linearized parameters for different clusters.

**Lemma C.2.** *If users $i$ and $l$ belong to different clusters, then we have that*

$$\sqrt{m_{NN}} \|\boldsymbol{\theta}_{f,i} - \boldsymbol{\theta}_{f,l}\| \geq \gamma'.$$

*Proof.* To begin with, Lemma C.1 tells us that

$$|f_i(\boldsymbol{x}) - f_l(\boldsymbol{x})| = |\langle g(\boldsymbol{x}; \boldsymbol{\theta}_0), \boldsymbol{\theta}_{f,i} - \boldsymbol{\theta}_{f,l}\rangle| \leq \|g(\boldsymbol{x}; \boldsymbol{\theta}_0)\| \|\boldsymbol{\theta}_{f,i} - \boldsymbol{\theta}_{f,l}\|. \tag{67}$$

This leads to

$$\|\boldsymbol{\theta}_{f,i} - \boldsymbol{\theta}_{f,l}\| \geq \frac{|f_i(\boldsymbol{x}) - f_l(\boldsymbol{x})|}{\|g(\boldsymbol{x}; \boldsymbol{\theta}_0)\|} \geq \frac{\gamma'}{\sqrt{m_{\text{NN}}}}, \tag{68}$$

in which we have made use of Assumption 2.7 and our assumption that $\frac{1}{m_{\text{NN}}}\langle g(\boldsymbol{x}; \boldsymbol{\theta}_0), g(\boldsymbol{x}; \boldsymbol{\theta}_0)\rangle \leq 1$ in the last inequality. This completes the proof. $\qquad\square$

The following lemma shows that for every user, the output of the NN trained using its own local data can be approximated by a linear function.

**Lemma C.3.** *Let $\varepsilon'_{m_{NN},t} \triangleq C_2 m_{NN}^{-1/6} \sqrt{\log m_{NN}} L^3 \left(\frac{t}{\lambda}\right)^{4/3}$ where $C_2 > 0$ is an absolute constant. Then*

$$|\langle g(\boldsymbol{x}; \boldsymbol{\theta}_0), \hat{\boldsymbol{\theta}}_{i,t} - \boldsymbol{\theta}_0\rangle - h(\boldsymbol{x}; \hat{\boldsymbol{\theta}}_{i,t})| \leq \varepsilon'_{m_{\text{NN}},t}, \quad \forall t \in [T], \boldsymbol{x}, \boldsymbol{x}' \in \mathcal{X}_t.$$

*Proof.* This lemma can be proved following a similar line of proof as Lemma 1 from Verma et al. (2024). Here the $t$ in $\varepsilon'_{m_{\text{NN}},t}$ can in fact be replaced by $T_{i,t} \leq t$, however, we have simply used its upper bound $t$ for simplicity. $\qquad\square$

**Lemma C.4.** *Let $\beta_T \triangleq \frac{1}{\kappa_\mu}\sqrt{\tilde{d} + 2\log(u/\delta)}$. Assuming that the conditions on $m_{NN}$ from Equation (66) are satisfied. With probability of at least $1 - \delta$, we have that*

$$\sqrt{m_{NN}} \left\|\boldsymbol{\theta}_{f,i} - \hat{\boldsymbol{\theta}}_{i,t}\right\|_2 \leq \frac{\beta_T + B\sqrt{\frac{\lambda}{\kappa_\mu}} + 1}{\sqrt{\lambda_{\min}(\boldsymbol{V}_{i,t-1})}}, \qquad \forall t \in [T].$$

*where $\boldsymbol{V}_{i,t-1} = \frac{\lambda}{\kappa_\mu}\boldsymbol{I} + \sum_{\substack{s \in [t-1] \\ i_s = i}} (\phi(\boldsymbol{x}_{s,1}) - \phi(\boldsymbol{x}_{s,2}))(\phi(\boldsymbol{x}_{s,1}) - \phi(\boldsymbol{x}_{s,2}))^\top$, $\phi(\boldsymbol{x}) = \frac{1}{\sqrt{m_{NN}}} g(\boldsymbol{x}; \boldsymbol{\theta}_0)$, and $T_{i,t}$ denotes the number of rounds of seeing user $i$ in the first $t$ rounds.*

*Proof.* In iteration $t$, for any user $i \in \mathcal{U}$, the user leverages its current history of observations $\{(\boldsymbol{x}_{s,1}, \boldsymbol{x}_{s,2}, y_s)\}_{s \in [t-1], i_s = i}$ to train the NN by minimizing the loss function (equation (19)), to obtain the NN parameters $\hat{\boldsymbol{\theta}}_{i,t}$. Note that the NN has been trained when the most recent observation in $\{(\boldsymbol{x}_{s,1}, \boldsymbol{x}_{s,2}, y_s)\}_{s \in [t-1], i_s = i}$ was collected, i.e., the last time when user $i$ was encountered. Of note, according to Lemma C.1, the latent reward function of user $i$ can be expressed as $f_i(\boldsymbol{x}) = \langle g(\boldsymbol{x}; \boldsymbol{\theta}_0), \boldsymbol{\theta}_{f,i} - \boldsymbol{\theta}_0\rangle$. Therefore, from the perspective of each individual user $i$, the user is faced with a *neural dueling bandit* problem instance. As a result, we can modifying the proof of Lemma 6 from Verma et al. (2024) to show that with probability of at least $1 - \delta$,

$$\sqrt{m_{\text{NN}}} \left\|\boldsymbol{\theta}_{f,i} - \hat{\boldsymbol{\theta}}_{i,t}\right\|_{\boldsymbol{V}_{i,t-1}} \leq \beta_T + B\sqrt{\frac{\lambda}{\kappa_\mu}} + 1, \qquad \forall t \in [T], i \in \mathcal{U}.$$

Here in our definition of $\beta_T \triangleq \frac{1}{\kappa_\mu}\sqrt{\tilde{d} + 2\log(u/\delta)}$, we have replaced the error probability $\delta$ (from Verma et al. (2024)) by $\delta/u$ to account for the use of an extra union bound over all $u$ users.

This allows us to show that

$$
\begin{aligned}
\sqrt{m_{\mathrm{NN}}} \left\| \boldsymbol{\theta}_{f,i} - \hat{\boldsymbol{\theta}}_{i,t} \right\|_2 &\le \frac{\sqrt{m_{\mathrm{NN}}} \left\| \boldsymbol{\theta}_{f,i} - \hat{\boldsymbol{\theta}}_{i,t} \right\|_{\boldsymbol{V}_{i,t-1}}}{\sqrt{\lambda_{\min}(\boldsymbol{V}_{i,t-1})}} \\
&\le \frac{\beta_T + B\sqrt{\frac{\lambda}{\kappa_\mu}} + 1}{\sqrt{\lambda_{\min}(\boldsymbol{V}_{i,t-1})}}
\end{aligned}
\tag{69}
$$

This completes the proof. $\qquad\square$

**Lemma C.5.** *With the carefully designed edge deletion rule in Algorithm 2, after*

$$
\begin{aligned}
T_0 &\triangleq 16u \log(\frac{u}{\delta}) + 4u \max\left\{ \frac{32\left(\widetilde{d} + 2\log(u/\delta)\right)}{\widetilde{\lambda}_x \gamma^2 \kappa_\mu^2}, \frac{16}{\widetilde{\lambda}_x^2} \log(\frac{24udm^2(L-1)}{\widetilde{\lambda}_x^2 \delta}) \right\} \\
&= O\left( u\left( \frac{\widetilde{d}}{\kappa_\mu^2 \widetilde{\lambda}_x \gamma^2} + \frac{1}{\widetilde{\lambda}_x^2} \right) \log(\frac{1}{\delta}) \right),
\end{aligned}
$$

*rounds, with probability at least $1 - 3\delta$ for some $\delta \in (0, \frac{1}{3})$, CONDB can cluster all the users correctly.*

*Proof.* Recall that we use $p = dm_{\mathrm{NN}} + m_{\mathrm{NN}}^2(L-1) + m_{\mathrm{NN}}$ to denote the total number of parameters of the NN. Similar to the proof of Lemma B.2, with the item regularity assumption stated in Assumption 2.4, Lemma J.1 in (Wang et al., 2024a), together with Lemma 7 in (Li & Zhang, 2018) (note that when using these technical results, we use $g(\boldsymbol{x}; \boldsymbol{\theta})/\sqrt{m_{\mathrm{NN}}}$ as the feature vector to replace the original feature vector of $\boldsymbol{x}$), and applying a union bound, with probability at least $1 - \delta$, for all $i \in \mathcal{U}$, at any $t$ such that $T_{i,t} \ge \frac{16}{\widetilde{\lambda}_x^2} \log(\frac{8up}{\widetilde{\lambda}_x^2 \delta})$, we have:

$$
\lambda_{\min}(\boldsymbol{V}_{i,t}) \ge 2\widetilde{\lambda}_x T_{i,t} .
\tag{70}
$$

Note that compared with the proof of B.2, in the lower bound on $T_{i,t}$ here, we have replaced the dimension $d$ by $p$. This has led to a logarithmic dependence on the width $m_{\mathrm{NN}}$ of the NN. To simplify the exposition, using the fact that $p \ge 3dm_{\mathrm{NN}}^2(L-1)$, we replace this condition on $T_{i,t}$ by a slightly stricter condition: $T_{i,t} \ge \frac{16}{\widetilde{\lambda}_x^2} \log(\frac{8u \times 3dm_{\mathrm{NN}}^2(L-1)}{\widetilde{\lambda}_x^2 \delta}) = \frac{16}{\widetilde{\lambda}_x^2} \log(\frac{24udm_{\mathrm{NN}}^2(L-1)}{\widetilde{\lambda}_x^2 \delta})$.

Then, together with Lemma C.4, we have: if $T_{i,t} \ge \frac{16}{\widetilde{\lambda}_x^2} \log(\frac{8u \times 3dm_{\mathrm{NN}}^2(L-1)}{\widetilde{\lambda}_x^2 \delta})$, then with probability $\ge 1 - 2\delta$, we have:

$$
\sqrt{m_{\mathrm{NN}}} \left\| \hat{\boldsymbol{\theta}}_{i,t} - \boldsymbol{\theta}^{j(i)} \right\| \le \frac{\beta_T + B\sqrt{\frac{\lambda}{\kappa_\mu}} + 1}{\sqrt{\lambda_{\min}(\boldsymbol{V}_{i,t-1})}} \le \frac{\beta_T + B\sqrt{\frac{\lambda}{\kappa_\mu}} + 1}{\sqrt{2\widetilde{\lambda}_x T_{i,t}}} .
$$

Now, let

$$
\frac{\beta_T + B\sqrt{\frac{\lambda}{\kappa_\mu}} + 1}{\sqrt{2\widetilde{\lambda}_x T_{i,t}}} < \frac{\gamma}{4} ,
\tag{71}
$$

Note that in Algorithm 2, we have defined the funciton $f$ as

$$
f(T_{i,t}) \triangleq \frac{\beta_T + B\sqrt{\frac{\lambda}{\kappa_\mu}} + 1}{\sqrt{2\widetilde{\lambda}_x T_{i,t}}}
\tag{72}
$$

This immediately leads to

$$
\sqrt{m_{\mathrm{NN}}} \left\| \hat{\boldsymbol{\theta}}_{i,t} - \boldsymbol{\theta}^{j(i)} \right\| \le f(T_{i,t}) < \frac{\gamma}{4}.
\tag{73}
$$

For simplicity, now let $B\sqrt{\frac{\lambda}{\kappa_\mu}} + 1 \le \beta_T$ which is typically satisfied. This allows us to show that

$$T_{i,t} > \frac{32\beta_T^2}{\widetilde{\lambda}_x \gamma^2} = \frac{32\left(\frac{1}{\kappa_\mu}\sqrt{\widetilde{d} + 2\log(u/\delta)}\right)^2}{\widetilde{\lambda}_x \gamma^2} = \frac{32\left(\widetilde{d} + 2\log(u/\delta)\right)}{\widetilde{\lambda}_x \gamma^2 \kappa_\mu^2}. \tag{74}$$

Combining both conditions on $T_{i,t}$ together, we have that

$$T_{i,t} \ge \max\left\{\frac{32\left(\widetilde{d} + 2\log(u/\delta)\right)}{\widetilde{\lambda}_x \gamma^2 \kappa_\mu^2}, \frac{16}{\widetilde{\lambda}_x^2}\log(\frac{24udm_{\mathrm{NN}}^2(L-1)}{\widetilde{\lambda}_x^2 \delta})\right\} \tag{75}$$

By Lemma 8 in (Li & Zhang, 2018) and Assumption 2.3 of user arrival uniformness, we have that for all

$$T_0 \triangleq 16u\log(\frac{u}{\delta}) + 4u\max\left\{\frac{32\left(\widetilde{d} + 2\log(u/\delta)\right)}{\widetilde{\lambda}_x \gamma^2 \kappa_\mu^2}, \frac{16}{\widetilde{\lambda}_x^2}\log(\frac{24udm_{\mathrm{NN}}^2(L-1)}{\widetilde{\lambda}_x^2 \delta})\right\}$$

$$= O\left(u\left(\frac{\widetilde{d}}{\kappa_\mu^2 \widetilde{\lambda}_x \gamma^2} + \frac{1}{\widetilde{\lambda}_x^2}\right)\log(\frac{1}{\delta})\right),$$

the condition in Eq.(74) is satisfied with probability at least $1 - \delta$.

Therefore we have that for all $t \ge T_0$, with probability $\ge 1 - 3\delta$:

$$\sqrt{m_{\mathrm{NN}}}\left\|\hat{\boldsymbol{\theta}}_{i,t} - \boldsymbol{\theta}^{j(i)}\right\|_2 < \frac{\gamma}{4}, \forall i \in \mathcal{U}. \tag{76}$$

Finally, we show that as long as the condition $\sqrt{m_{\mathrm{NN}}}\left\|\hat{\boldsymbol{\theta}}_{i,t} - \boldsymbol{\theta}^{j(i)}\right\|_2 < \frac{\gamma}{4}, \forall i \in \mathcal{U}$, our algorithm can cluster all the users correctly.

First, we show that when the edge $(i, l)$ is deleted, user $i$ and user $j$ must belong to different *ground-truth clusters*, i.e., $\|\boldsymbol{\theta}_{f,i} - \boldsymbol{\theta}_{f,l}\|_2 > 0$. This is because by the deletion rule of the algorithm, the concentration bound, and triangle inequality

$$\sqrt{m_{\mathrm{NN}}}\|\boldsymbol{\theta}_{f,i} - \boldsymbol{\theta}_{f,l}\|_2 = \sqrt{m_{\mathrm{NN}}}\left\|\boldsymbol{\theta}^{j(i)} - \boldsymbol{\theta}^{j(l)}\right\|_2$$

$$\ge \sqrt{m_{\mathrm{NN}}}\left\|\hat{\boldsymbol{\theta}}_{i,t} - \hat{\boldsymbol{\theta}}_{l,t}\right\|_2 - \sqrt{m_{\mathrm{NN}}}\left\|\boldsymbol{\theta}^{j(l)} - \hat{\boldsymbol{\theta}}_{l,t}\right\|_2 - \sqrt{m_{\mathrm{NN}}}\left\|\boldsymbol{\theta}^{j(i)} - \hat{\boldsymbol{\theta}}_{i,t}\right\|_2$$

$$\ge \sqrt{m_{\mathrm{NN}}}\left\|\hat{\boldsymbol{\theta}}_{i,t} - \hat{\boldsymbol{\theta}}_{l,t}\right\|_2 - f(T_{i,t}) - f(T_{l,t}) > 0. \tag{77}$$

Second, we can show that if $|f_i(\boldsymbol{x}) - f_l(\boldsymbol{x})| \ge \gamma', \forall \boldsymbol{x} \in \mathcal{X}$, meaning that user $i$ and user $l$ are not in the same *ground-truth cluster*, CONDB will delete the edge $(i, l)$ after $T_0$. Note that when user $i$ and user $l$ are not in the same *ground-truth cluster*, Lemma C.2 tells us that $\sqrt{m_{\mathrm{NN}}}\|\boldsymbol{\theta}_{f,i} - \boldsymbol{\theta}_{f,l}\| \ge \gamma'$. Then we have that

$$\sqrt{m_{\mathrm{NN}}}\left\|\hat{\boldsymbol{\theta}}_{i,t} - \hat{\boldsymbol{\theta}}_{l,t}\right\| \ge \sqrt{m_{\mathrm{NN}}}\|\boldsymbol{\theta}_{f,i} - \boldsymbol{\theta}_{f,l}\| - \sqrt{m_{\mathrm{NN}}}\left\|\hat{\boldsymbol{\theta}}_{i,t} - \boldsymbol{\theta}^{j(i)}\right\|_2 - \sqrt{m_{\mathrm{NN}}}\left\|\hat{\boldsymbol{\theta}}_{l,t} - \boldsymbol{\theta}^{j(l)}\right\|_2$$

$$> \gamma - \frac{\gamma}{4} - \frac{\gamma}{4}$$

$$= \frac{\gamma}{2} > f(T_{i,t}) + f(T_{l,t}), \tag{78}$$

which will trigger the edge deletion rule to delete edge $(i, l)$. This completes the proof. $\square$

Then, we prove the following lemmas for the cluster-based statistics.

**Lemma C.6.** *Assuming that the conditions on $m$ from Equation (66) are satisfied. With probability at least $1 - 4\delta$ for some $\delta \in (0, 1/4)$, at any $t \geq T_0$:*

$$\sqrt{m_{NN}} \left\| \boldsymbol{\theta}_{f,i_t} - \overline{\boldsymbol{\theta}}_t \right\|_{\boldsymbol{V}_{t-1}} \leq \beta_T + B\sqrt{\frac{\lambda}{\kappa_\mu}} + 1, \qquad \forall t \in [T].$$

*Proof.* To begin with, note that by Lemma C.5, we have that with probability of at least $1 - 3\delta$, all users are clustered correctly, i.e., $\overline{C}_t = C_{j(i_t)}, \forall t \geq T_0$. Note that according to our Algorithm 2, in iteration $t$, we select the pair of arms using all the data collected by all users in cluster $\overline{C}_t$. That is, $\overline{\boldsymbol{\theta}}_t$ represents the NN parameters trained using the data from all users in the cluster $\overline{C}_t$ (i.e., $\{(\boldsymbol{x}_{s,1}, \boldsymbol{x}_{s,2}, y_s)\}_{s \in [t-1], i_s \in \overline{C}_t}$), and $\boldsymbol{V}_t$ also contains the data from all users in this cluster $\overline{C}_t$. Therefore, in iteration $t$, we are effectively following a neural dueling bandit algorithm using $\{(\boldsymbol{x}_{s,1}, \boldsymbol{x}_{s,2}, y_s)\}_{s \in [t-1], i_s \in \overline{C}_t}$ as the current observation history. This allows us to leverage the proof of Lemma 6 from Verma et al. (2024) to complete the proof. $\square$

**Lemma C.7.** *Let $\varepsilon'_{m_{NN},t} \triangleq C_2 m_{NN}^{-1/6} \sqrt{\log m_{NN}} L^3 \left(\frac{t}{\lambda}\right)^{4/3}$ where $C_2 > 0$ is an absolute constant. Then*

$$|\langle g(\boldsymbol{x}; \boldsymbol{\theta}_0) - g(\boldsymbol{x}'; \boldsymbol{\theta}_0), \overline{\boldsymbol{\theta}}_t - \boldsymbol{\theta}_0 \rangle - (h(\boldsymbol{x}; \overline{\boldsymbol{\theta}}_t) - h(\boldsymbol{x}'; \overline{\boldsymbol{\theta}}_t))| \leq 2\varepsilon'_{m_{NN},t}, \ \ \forall t \in [T], \boldsymbol{x}, \boldsymbol{x}' \in \mathcal{X}_t.$$

*Proof.* This lemma can be proved following a similar line of proof as Lemma 1 from Verma et al. (2024). $\square$

**Lemma C.8.** *Let $\delta \in (0, 1)$, $\varepsilon'_{m_{NN},t} \doteq C_2 m_{NN}^{-1/6} \sqrt{\log m_{NN}} L^3 \left(\frac{t}{\lambda}\right)^{4/3}$ for some absolute constant $C_2 > 0$. As long as $m_{NN} \geq poly(T, L, K, u, 1/\kappa_\mu, L_\mu, 1/\lambda_0, 1/\lambda, \log(1/\delta))$, then with probability of at least $1 - \delta$, at any $t \geq T_0$,*

$$| [f_{i_t}(\boldsymbol{x}) - f_{i_t}(\boldsymbol{x}')] - [h(\boldsymbol{x}; \overline{\boldsymbol{\theta}}_t) - h(\boldsymbol{x}'; \overline{\boldsymbol{\theta}}_t)] | \leq \nu_T \sigma_{t-1}(\boldsymbol{x}, \boldsymbol{x}') + 2\varepsilon'_{m_{NN},t},$$

*for all $\boldsymbol{x}, \boldsymbol{x}' \in \mathcal{X}_t, t \in [T]$.*

*Proof.* Denote $\phi(\boldsymbol{x}) = \frac{1}{\sqrt{m_{NN}}} g(\boldsymbol{x}; \boldsymbol{\theta}_0)$. Recall that Lemma C.1 tells us that $f_{i_t}(\boldsymbol{x}) = \langle g(\boldsymbol{x}; \boldsymbol{\theta}_0), \boldsymbol{\theta}_{f,i_t} - \boldsymbol{\theta}_0 \rangle = \langle \phi(\boldsymbol{x}), \boldsymbol{\theta}_{f,i_t} - \boldsymbol{\theta}_0 \rangle$ for all $\boldsymbol{x} \in \mathcal{X}_t, t \in [T]$. To begin with, for all $\boldsymbol{x}, \boldsymbol{x}' \in \mathcal{X}_t, t \in [T]$ we have that

$$
\begin{aligned}
&|f_{i_t}(\boldsymbol{x}) - f_{i_t}(\boldsymbol{x}') - \langle g(\boldsymbol{x}; \boldsymbol{\theta}_0) - g(\boldsymbol{x}'; \boldsymbol{\theta}_0), \overline{\boldsymbol{\theta}}_t - \boldsymbol{\theta}_0 \rangle| \\
&= |\langle g(\boldsymbol{x}; \boldsymbol{\theta}_0) - g(\boldsymbol{x}'; \boldsymbol{\theta}_0), \boldsymbol{\theta}_{f,i_t} - \boldsymbol{\theta}_0 \rangle - \langle g(\boldsymbol{x}; \boldsymbol{\theta}_0) - g(\boldsymbol{x}'; \boldsymbol{\theta}_0), \overline{\boldsymbol{\theta}}_t - \boldsymbol{\theta}_0 \rangle| \\
&= |\langle g(\boldsymbol{x}; \boldsymbol{\theta}_0) - g(\boldsymbol{x}'; \boldsymbol{\theta}_0), \boldsymbol{\theta}_{f,i_t} - \overline{\boldsymbol{\theta}}_t \rangle| \\
&= |\langle \phi(\boldsymbol{x}) - \phi(\boldsymbol{x}'), \sqrt{m_{NN}} (\boldsymbol{\theta}_{f,i_t} - \overline{\boldsymbol{\theta}}_t) \rangle| \\
&\leq \|(\phi(\boldsymbol{x}) - \phi(\boldsymbol{x}'))\|_{\boldsymbol{V}_{t-1}^{-1}} \sqrt{m_{NN}} \left\| \boldsymbol{\theta}_{f,i_t} - \overline{\boldsymbol{\theta}}_t \right\|_{\boldsymbol{V}_{t-1}} \\
&\leq \|(\phi(\boldsymbol{x}) - \phi(\boldsymbol{x}'))\|_{\boldsymbol{V}_{t-1}^{-1}} \left( \beta_T + B\sqrt{\frac{\lambda}{\kappa_\mu}} + 1 \right),
\end{aligned}
\tag{79}
$$

in which we have used Lemma C.6 in the last inequality. Now making use of the equation above and Lemma C.7, we have that

$$
\begin{aligned}
&|f_{i_t}(\boldsymbol{x}) - f_{i_t}(\boldsymbol{x}') - (h(\boldsymbol{x}; \boldsymbol{\theta}_t) - h(\boldsymbol{x}'; \boldsymbol{\theta}_t))| \\
&= |f_{i_t}(\boldsymbol{x}) - f_{i_t}(\boldsymbol{x}') - \langle g(\boldsymbol{x}; \boldsymbol{\theta}_0) - g(\boldsymbol{x}'; \boldsymbol{\theta}_0), \overline{\boldsymbol{\theta}}_t - \boldsymbol{\theta}_0 \rangle \\
&\qquad + \langle g(\boldsymbol{x}; \boldsymbol{\theta}_0) - g(\boldsymbol{x}'; \boldsymbol{\theta}_0), \overline{\boldsymbol{\theta}}_t - \boldsymbol{\theta}_0 \rangle - (h(\boldsymbol{x}; \overline{\boldsymbol{\theta}}_t) - h(\boldsymbol{x}'; \overline{\boldsymbol{\theta}}_t))| \\
&\leq |f_{i_t}(\boldsymbol{x}) - f_{i_t}(\boldsymbol{x}') - \langle g(\boldsymbol{x}; \boldsymbol{\theta}_0) - g(\boldsymbol{x}'; \boldsymbol{\theta}_0), \overline{\boldsymbol{\theta}}_t - \boldsymbol{\theta}_0 \rangle| \\
&\qquad + |\langle g(\boldsymbol{x}; \boldsymbol{\theta}_0) - g(\boldsymbol{x}'; \boldsymbol{\theta}_0), \overline{\boldsymbol{\theta}}_t - \boldsymbol{\theta}_0 \rangle - (h(\boldsymbol{x}; \overline{\boldsymbol{\theta}}_t) - h(\boldsymbol{x}'; \overline{\boldsymbol{\theta}}_t))| \\
&\leq \left\| \frac{1}{\sqrt{m_{NN}}} (\phi(\boldsymbol{x}) - \phi(\boldsymbol{x}')) \right\|_{\boldsymbol{V}_{t-1}^{-1}} \left( \beta_T + B\sqrt{\frac{\lambda}{\kappa_\mu}} + 1 \right) + 2\varepsilon'_{m_{NN},t}.
\end{aligned}
\tag{80}
$$

This completes the proof. $\square$

We also prove the following lemma to upper bound the summation of squared norms which will be used in proving the final regret bound.

**Lemma C.9.** *With probability at least $1 - 4\delta$, we have*

$$\sum_{t=T_0}^{T} \mathbb{I}\{i_t \in C_j\} \|\phi(\boldsymbol{x}_{t,1}) - \phi(\boldsymbol{x}_{t,2})\|_{\boldsymbol{V}_{t-1}^{-1}}^2 \leq 16\widetilde{d}, \forall j \in [m],$$

where $\mathbb{I}$ denotes the indicator function.

*Proof.* We denote $\widetilde{\phi}_t = \phi(\boldsymbol{x}_{t,1}) - \phi(\boldsymbol{x}_{t,2})$. Note that we have defined $\phi(\boldsymbol{x}) = \frac{1}{\sqrt{m_{\text{NN}}}} g(\boldsymbol{x}; \boldsymbol{\theta}_0)$. Here we assume that $\|\phi(\boldsymbol{x}_{t,1}) - \phi(\boldsymbol{x}_{t,2})\|_2 = \frac{1}{\sqrt{m_{\text{NN}}}} \|g(\boldsymbol{x}_{t,1}; \boldsymbol{\theta}_0) - g(\boldsymbol{x}_{t,2}; \boldsymbol{\theta}_0)\|_2 \leq 2$. Replacing 2 by an absolute constant $c_0$ would only change the final regret bound by a constant factor, so we omit it for simplicity.

It is easy to verify that $\boldsymbol{V}_{t-1} \succeq \frac{\lambda}{\kappa_\mu} I$ and hence $\boldsymbol{V}_{t-1}^{-1} \preceq \frac{\kappa_\mu}{\lambda} I$. Therefore, we have that $\left\|\widetilde{\phi}_t\right\|_{\boldsymbol{V}_{t-1}^{-1}}^2 \leq \frac{\kappa_\mu}{\lambda} \left\|\widetilde{\phi}_t\right\|_2^2 \leq \frac{4\kappa_\mu}{\lambda}$. We choose $\lambda$ such that $\frac{4\kappa_\mu}{\lambda} \leq 1$, which ensures that $\left\|\widetilde{\phi}_t\right\|_{\boldsymbol{V}_{t-1}^{-1}}^2 \leq 1$. Our proof here mostly follows from Lemma 11 of (Abbasi-Yadkori et al., 2011) and Lemma J.2 of (Wang et al., 2024a). To begin with, note that $x \leq 2\log(1+x)$ for $x \in [0,1]$. Denote $\boldsymbol{V}_{t,j} = \sum_{\substack{s \in [t-1]: \\ i_s \in C_j}} \widetilde{\phi}_s \widetilde{\phi}_s^\top + \frac{\lambda}{\kappa_\mu} \mathbf{I}$. Then we have that

$$
\begin{aligned}
\sum_{t=T_0}^{T} \mathbb{I}\{i_t \in C_j\} \left\|\widetilde{\phi}_t\right\|_{\boldsymbol{V}_{t-1}^{-1}}^2 &\leq \sum_{t=T_0}^{T} 2\log\left(1 + \mathbb{I}\{i_t \in C_j\} \left\|\widetilde{\phi}_t\right\|_{\boldsymbol{V}_{t-1}^{-1}}^2\right) \\
&\leq 16\log\det\left(\frac{\kappa_\mu}{\lambda} \mathbf{H}' + \mathbf{I}\right) \\
&\triangleq 16\widetilde{d}.
\end{aligned}
\tag{81}
$$

The second inequality follows from the proof in Section A.3 from Verma et al. (2024). This completes the proof. $\qquad\square$

Now we are ready to prove Theorem 4.2. To begin with, we have that $R_T = \sum_{t=1}^{T} r_t \leq T_0 + \sum_{t=T_0}^{T} r_t$.

Then, we only need to upper-bound the regret after $T_0$. By Lemma C.5, we know that with probability at least $1 - 4\delta$, the algorithm can cluster all the users correctly, $\overline{C}_t = C_{j(i_t)}$, and the statements of all the above lemmas hold. We have that for any $t \geq T_0$:

To simplify exposion here, we denote $\beta'_T \triangleq \beta_T + B\sqrt{\lambda/\kappa_\mu} + 1$.

$$
\begin{aligned}
r_t &= f_{i_t}(\boldsymbol{x}_t^*) - f_{i_t}(\boldsymbol{x}_{t,1}) + f_{i_t}(\boldsymbol{x}_t^*) - f_{i_t}(x_{t,2}) \\
&\overset{(a)}{\leq} \langle g(\boldsymbol{x}_t^*; \boldsymbol{\theta}_0) - g(\boldsymbol{x}_{t,1}; \boldsymbol{\theta}_0), \overline{\boldsymbol{\theta}}_t - \boldsymbol{\theta}_0 \rangle + \beta'_T \|\phi(\boldsymbol{x}_t^*) - \phi(\boldsymbol{x}_{t,1})\|_{\boldsymbol{V}_{t-1}^{-1}} + \\
&\quad \langle g(\boldsymbol{x}_t^*; \boldsymbol{\theta}_0) - g(\boldsymbol{x}_{t,2}; \boldsymbol{\theta}_0), \overline{\boldsymbol{\theta}}_t - \boldsymbol{\theta}_0 \rangle + \beta'_T \|\phi(\boldsymbol{x}_t^*) - \phi(\boldsymbol{x}_{t,2})\|_{\boldsymbol{V}_{t-1}^{-1}} \\
&= \langle g(\boldsymbol{x}_t^*; \boldsymbol{\theta}_0) - g(\boldsymbol{x}_{t,1}; \boldsymbol{\theta}_0), \overline{\boldsymbol{\theta}}_t - \boldsymbol{\theta}_0 \rangle + \beta'_T \|\phi(\boldsymbol{x}_t^*) - \phi(\boldsymbol{x}_{t,1})\|_{\boldsymbol{V}_{t-1}^{-1}} + \\
&\quad \langle g(\boldsymbol{x}_t^*; \boldsymbol{\theta}_0) - g(\boldsymbol{x}_{t,1}; \boldsymbol{\theta}_0), \overline{\boldsymbol{\theta}}_t - \boldsymbol{\theta}_0 \rangle + \langle g(\boldsymbol{x}_{t,1}; \boldsymbol{\theta}_0) - g(\boldsymbol{x}_{t,2}; \boldsymbol{\theta}_0), \overline{\boldsymbol{\theta}}_t - \boldsymbol{\theta}_0 \rangle + \\
&\quad \beta'_T \|\phi(\boldsymbol{x}_t^*) - \phi(\boldsymbol{x}_{t,1}) + \phi(\boldsymbol{x}_{t,1}) - \phi(\boldsymbol{x}_{t,2})\|_{\boldsymbol{V}_{t-1}^{-1}} \\
&\overset{(b)}{\leq} 2\langle g(\boldsymbol{x}_t^*; \boldsymbol{\theta}_0) - g(\boldsymbol{x}_{t,1}; \boldsymbol{\theta}_0), \overline{\boldsymbol{\theta}}_t - \boldsymbol{\theta}_0 \rangle + 2\beta'_T \|\phi(\boldsymbol{x}_t^*) - \phi(\boldsymbol{x}_{t,1})\|_{\boldsymbol{V}_{t-1}^{-1}} + \\
&\quad \langle g(\boldsymbol{x}_{t,1}; \boldsymbol{\theta}_0) - g(\boldsymbol{x}_{t,2}; \boldsymbol{\theta}_0), \overline{\boldsymbol{\theta}}_t - \boldsymbol{\theta}_0 \rangle + \beta'_T \|\phi(\boldsymbol{x}_{t,1}) - \phi(\boldsymbol{x}_{t,2})\|_{\boldsymbol{V}_{t-1}^{-1}} \\
&\overset{(c)}{\leq} 2h(\boldsymbol{x}_t^*; \overline{\boldsymbol{\theta}}_t) - 2h(\boldsymbol{x}_{t,1}; \overline{\boldsymbol{\theta}}_t) + 4\varepsilon'_{m_{\text{NN}},t} + 2\beta'_T \|\phi(\boldsymbol{x}_t^*) - \phi(\boldsymbol{x}_{t,1})\|_{\boldsymbol{V}_{t-1}^{-1}} + \\
&\quad h(\boldsymbol{x}_{t,1}; \overline{\boldsymbol{\theta}}_t) - h(\boldsymbol{x}_{t,2}; \overline{\boldsymbol{\theta}}_t) + 2\varepsilon'_{m_{\text{NN}},t} + \beta'_T \|\phi(\boldsymbol{x}_{t,1}) - \phi(\boldsymbol{x}_{t,2})\|_{\boldsymbol{V}_{t-1}^{-1}} \\
&\overset{(d)}{\leq} 2h(\boldsymbol{x}_{t,2}; \overline{\boldsymbol{\theta}}_t) - 2h(\boldsymbol{x}_{t,1}; \overline{\boldsymbol{\theta}}_t) + 2\beta'_T \|\phi(\boldsymbol{x}_{t,2}) - \phi(\boldsymbol{x}_{t,1})\|_{\boldsymbol{V}_{t-1}^{-1}} + \\
&\quad h(\boldsymbol{x}_{t,1}; \overline{\boldsymbol{\theta}}_t) - h(\boldsymbol{x}_{t,2}; \overline{\boldsymbol{\theta}}_t) + 6\varepsilon'_{m_{\text{NN}},t} + \beta'_T \|\phi(\boldsymbol{x}_{t,1}) - \phi(\boldsymbol{x}_{t,2})\|_{\boldsymbol{V}_{t-1}^{-1}} \\
&= h(\boldsymbol{x}_{t,2}; \overline{\boldsymbol{\theta}}_t) - h(\boldsymbol{x}_{t,1}; \overline{\boldsymbol{\theta}}_t) + 3\beta'_T \|\phi(\boldsymbol{x}_{t,1}) - \phi(\boldsymbol{x}_{t,2})\|_{\boldsymbol{V}_{t-1}^{-1}} + 6\varepsilon'_{m_{\text{NN}},t} \\
&\overset{(e)}{\leq} 3\beta'_T \|\phi(\boldsymbol{x}_{t,1}) - \phi(\boldsymbol{x}_{t,2})\|_{\boldsymbol{V}_{t-1}^{-1}} + 6\varepsilon'_{m_{\text{NN}},t}
\end{aligned}
\tag{82}
$$

Step $(a)$ follows from Equation 79, step $(b)$ results from the triangle inequality, step $(c)$ has made use of Lemma C.7. Step $(d)$ follows from the way in which we choose the second arm $\boldsymbol{x}_{t,2}$: $\boldsymbol{x}_{t,2} = \arg\max_{\boldsymbol{x} \in \mathcal{X}_t} h(\boldsymbol{x}; \overline{\boldsymbol{\theta}}_t) + \left(\beta_T + B\sqrt{\frac{\lambda}{\kappa_\mu}} + 1\right) \|(\phi(\boldsymbol{x}) - \phi(\boldsymbol{x}_{t,1}))\|_{\boldsymbol{V}_{t-1}^{-1}}$. Step $(e)$ results from the way in which we select the first arm: $\boldsymbol{x}_{t,1} = \arg\max_{\boldsymbol{x} \in \mathcal{X}_t} h(\boldsymbol{x}; \overline{\boldsymbol{\theta}}_t)$.

Then we have

$$
\begin{aligned}
\sum_{t=T_0}^{T} r_t &\leq 3\beta'_T \sum_{t=T_0}^{T} \|\phi(\boldsymbol{x}_{t,1}) - \phi(\boldsymbol{x}_{t,2})\|_{\boldsymbol{V}_{t-1}^{-1}} + 6T\varepsilon'_{m_{\text{NN}},T} \\
&= 3\beta'_T \sum_{t=T_0}^{T} \sum_{j\in[m]} \mathbb{I}\{i_t \in C_j\} \|\phi(\boldsymbol{x}_{t,1}) - \phi(\boldsymbol{x}_{t,2})\|_{\boldsymbol{V}_{t-1}^{-1}} + 6T\varepsilon'_{m_{\text{NN}},T} \\
&\leq 3\beta'_T \sqrt{\sum_{t=T_0}^{T} \sum_{j\in[m]} \mathbb{I}\{i_t \in C_j\} \sum_{t=T_0}^{T} \sum_{j\in[m]} \mathbb{I}\{i_t \in C_j\} \|\phi(\boldsymbol{x}_{t,1}) - \phi(\boldsymbol{x}_{t,2})\|_{\boldsymbol{V}_{t-1}^{-1}}^2} + 6T\varepsilon'_{m_{\text{NN}},T} \\
&\leq 3\beta'_T \sqrt{T \cdot m \cdot 16\widetilde{d}} + 6T\varepsilon'_{m_{\text{NN}},T} \tag{83} \\
&\leq 12\beta'_T \sqrt{T \cdot m \cdot \widetilde{d}} + 6T\varepsilon'_{m_{\text{NN}},T}, \tag{84}
\end{aligned}
$$

where in the second inequality we use the Cauchy-Swarchz inequality, and in the last step we use $\sum_{t=T_0}^{T} \sum_{j\in[m]} \mathbb{I}\{i_t \in C_j\} \leq T$ and Lemma C.9. It can be easily verified that as long as the conditions on $m$ specified in Equation (66) are satisfied (i.e., as long as the NN is wide enough), we have that $6T\varepsilon'_{m_{\text{NN}},T} \leq 1$.

Recall that $\beta'_T \triangleq \beta_T + B\sqrt{\lambda/\kappa_\mu} + 1$ and $\beta_T \triangleq \frac{1}{\kappa_\mu}\sqrt{\widetilde{d} + 2\log(u/\delta)}$. Therefore, finally, we have with probability at least $1 - 4\delta$

$$
R_T \leq T_0 + 12(\beta_T + B\sqrt{\lambda/\kappa_\mu} + 1)\sqrt{T \cdot m \cdot \widetilde{d}} + 1
$$

$$\leq O\left(u(\frac{\widetilde{d}}{\kappa_\mu^2 \widetilde{\lambda}_x \gamma^2} + \frac{1}{\widetilde{\lambda}_x^2})\log T + \left(\frac{\sqrt{\widetilde{d}}}{\kappa_\mu} + B\sqrt{\frac{\lambda}{\kappa_\mu}}\right)\sqrt{\widetilde{d}mT}\right)$$

$$= O\left(\left(\frac{\sqrt{\widetilde{d}}}{\kappa_\mu} + B\sqrt{\frac{\lambda}{\kappa_\mu}}\right)\sqrt{\widetilde{d}mT}\right). \tag{85}$$

