# OpenReview forum: "Online Clustering of Dueling Bandits"
_ICML.cc/2025/Conference — ICML 2025 poster_

### Official Review · Reviewer_AU83 · 2025-03-14

**Overall Recommendation:** 3

**Summary:**

The work studies the integrated setting of clustering and dueling bandits. The clustering bandit setting assumes the arms are clustered where each cluster shares the same reward function. The dueling bandit setting assumes that the learner chooses two arms in each iteration and obtains the preference feedback as reward. The paper proposes algorithms for two variants of the problem, linear and neural contextual setting, together with regret bound analysis. Experiments under both synthetic and real-world data are conducted to verify the empirical effectiveness of the proposed approach.

**Claims And Evidence:**

- Originality: It is claimed that the work is the first to integrate the setting of clustering and dueling bandits. To my knowledge, the claim is true.

- Theoretical guarantee: The proposed approach is companioned with regret bound analysis with complete proofs. On the other hand, the optimality of the algorithm is not sufficiently justified, in particular w.r.t. the information on the clusters, such as cluster number.

- Experiments: In my view, the experimental results are solid.

**Essential References Not Discussed:**

NA

**Experimental Designs Or Analyses:**

The experimental part is solid.

**Methods And Evaluation Criteria:**

- The proposed algorithm utilizes relative standard techniques to handle clustering and dueling bandits. In my view, the method makes sense and is technically sound.

- The major performance metric is regret bound, which is a standard one for online learning.

**Other Comments Or Suggestions:**

- It would be better to further discuss the optimality of the order of $m$ in Theorem 4.1 and 4.2. In the current results, if $m$ is very large, such as achieving the order of $O(T)$, the regret bound would be significantly worse, which can be worse than without considering the clustering assumption and only assume the adversarial setting. So I wonder whether there is the room of improvement on the term of $m$.

- It would be better to introduce real-world datasets that natively follows the clustering and dueling assumptions.

**Other Strengths And Weaknesses:**

Strengths:

- The paper is technically sound. The proposed algorithm builds upon mature techniques. The technical proofs are complete.

- The paper is clearly written.

Weaknesses:

- The technical challenge to integrate the clustering and dueling bandits lacks clear explanations. It would be not satisfactory to integrate the two setting without discovering more in-depth technical insights.

- From practical perspective, it remains unclear why the proposed scenario is important in practice. In the introduction part, the example of LLM is discussed. While it remains unclear if the clustering assumption widely appears in real situations.

**Questions For Authors:**

1. It would be desirable to better highlight the technical challenges in algorithm design and theoretical analysis.

**Relation To Broader Scientific Literature:**

The work involves the clustering and dueling bandit learning scenarios, which have both received thorough studies. The work majorly studies the integration of both settings. From technical perspective, the paper utilizes relative standard techniques in algorithm design and theoretical analysis from the both scenarios.

**Theoretical Claims:**

I have checked the proofs, which is correct in my view.

---

> ### Author Rebuttal · Authors · 2025-04-01
>
> Thanks for the valuable advice. Our responses are as follows. We will add the discussions.
>
> **Q1:** Technical challenges:
>
> **A1:**
>
> **Novel Algorithm Design:**
> - Cluster Estimation under Dueling Feedback: Our setting does not allow direct use of prior cluster vector estimation methods in classical CB with numerical feedback. With numerical feedback, closed-form ridge regression enables simple updates of user-level covariance matrices and regressors, followed by cluster vector computation using aggregated statistics (e.g., Lines 6–8 in Algo.1 of Wang et al., 2024a). In contrast, dueling feedback lacks closed-form MLE, making direct adaptation infeasible. To address this, we maintain a history buffer $\mathcal{D}_t$ (e.g., Line 10, Algo.1) recording user indices, arm choices, and dueling feedback. We then filter data for users in inferred clusters and estimate cluster vectors via MLE (e.g., Line 6, Algo.1).
>
> - Carefully Designed Edge Deletion Thresholds: Threshold design under dueling feedback is more intricate than in absolute feedback settings and relies on refined calculations (Lemmas B.2 and C.5) to ensure correctness. In the neural setting, this is further complicated by the non-linear reward function. The required Cluster Separation assumption (Assumption 2.7) deviates from those in prior CB works, making its integration into the edge deletion rule for CONDB particularly challenging.
>
> **Novel Theoretical Analysis**:
> We highlight the main technical challenges for CONDB as an example:
> - Incorporating the novel Cluster Separation assumption (Assumption 2.7) for CONDB into the analytical framework of linear CB poses substantial difficulties. We address this by leveraging a linear approximation of the neural net as the analytical basis (Lemmas C.1 and C.2).
> - Deriving $T_0$ after which all users are correctly clustered (Lemma C.5), is particularly challenging due to the **non-linear and dueling nature of the reward**. Classical CB analysis bounds the L2 distance between true and estimated **linear parameters**, which is not applicable here. To overcome this, we use neural network linearization and rigorously analyze the resulting approximation error (Lemmas C.3, C.4, and C.5).
> - A key challenge in proving a sub-linear bound lies in the proper use of the effective dimension $\widetilde d$ (Eq.(16)). Direct extensions of Lemma B.5 from the COLDB (linear rewards) analysis would yield a bound linearly dependent on $p=O(m_{\text{NN}}^2)$, which becomes vacuous as $m_{\text{NN}}$ grows polynomially with $T$. To address this, we adopt techniques from neural bandits to reformulate Lemma B.5 for CONDB, allowing us to express the regret in terms of $\widetilde d$ (Lemma C.9).
>
> **Q2:** Setting Importance and LLM applications:
>
> **A2:**
> - Motivation: Leveraging user relations to accelerate learning is well-established in CB literature (e.g., Gentile et al., 2014), particularly in recommender systems where users naturally form groups [1]. Preference-based feedback is also more aligned with human behavior and common in applications like RLHF, motivating the use of user relations under dueling feedback.
> - Extension to More General Settings: Some works have explored similar-but-not-identical user preferences under numerical feedback (Ban & He, 2021b; Wang et al., 2024b). As the first to study preference feedback in CB, we focus on standard assumptions to address core challenges. Extending to those settings is left for future work.
> - LLM Applications: Our methods can directly extend recent work on neural dueling bandits for prompt optimization [2] to the multi-user setting.
>
> **Q3:** Optimality in $m$ and lower bounds:
>
> **A3:**
> - Following Wang et al. (2024a) and Saha (2021), we can derive a lower bound of $O(\sqrt{dmT})$ for linear case. Algo.1 achieves an upper bound of $O(d\sqrt{mT})$, which is tight up to $\sqrt d$—a common gap in linear bandits (e.g., LinUCB). Thus, our upper bound is optimal in $m$ for the linear case. For the neural case, lower bounds remain open even in the single-user setting. However, we hypothesize that our bound is also optimal in $m$ in the neural setting, as clustering naturally introduces a $\sqrt m$ scaling over the single-user case.
> - While adversarial bandit algorithms can achieve $O(\sqrt T)$ regret, **the regret definition differs**: adversarial regret compares to the best fixed arm, while stochastic regret compares to the optimal dynamic policy. Direct comparisons are not meaningful.
>
>
> **Q4:** Real-world data natively following clustering and dueling assumptions:
>
> **A4:** Our dataset and clustering setup follow standard practice in prior CB works using common benchmarks. We added a real-world dataset (Yelp), where COLDB continues to outperform baselines (see [figure](https://postimg.cc/QHS7rsbP)). We'll explore other datasets in future work.
>
> Reference
>
> [1] ClustKNN: a highly scalable hybrid model-& memory-based CF algorithm, KDD 2006.
>
> [2] Prompt Optimization with Human Feedback, 2024.

---

> > ### Comment · Reviewer_AU83 · 2025-04-04
> >
> > Thanks for the detailed responses, especially for the detailed explanations on the technical contributions in the theoretical analysis. I will further verify and take them into considerations for final evaluations.

---

> > > ### Author Response · Authors · 2025-04-05
> > >
> > > Thank you for your thoughtful feedback and for taking the time to acknowledge our detailed responses. We deeply appreciate your willingness to further verify our our explanations and consider them in the final evaluation. We hope that our responses could strengthen your confidence in our work.

---

### Official Review · Reviewer_epsw · 2025-03-14

**Overall Recommendation:** 3

**Summary:**

This paper primarily investigates dueling bandits in an online clustering setting, where clusters of arms share a reward function. It examines cases where the reward function is linear or modeled by a deep neural network. The authors propose algorithms with sub-linear regret bounds and demonstrate their effectiveness through experiments.

**Claims And Evidence:**

The claim of achieving sub-linear regret bounds is substantiated by theoretical proofs.

**Essential References Not Discussed:**

The references sufficiently cover the relevant literature.

**Experimental Designs Or Analyses:**

The experimental designs are reasonable, and the experimental results are promising.

**Methods And Evaluation Criteria:**

The proposed method maintains the clustering structure of users and estimates the reward function using information from users within the same cluster, which is appropriate for the clustering bandits problem.

**Other Comments Or Suggestions:**

I would suggest elaborating on the technical challenges involved in the algorithm design and its analysis beyond the application of known results from dueling bandits and clustering bandits.

**Other Strengths And Weaknesses:**

Strengths: The paper is well-organized, and the proposed algorithm is straightforward to comprehend. The authors demonstrate that the algorithm can achieve sub-linear regret bounds and provide clear intuition for the two terms in the regret bound. The experimental results further confirm the algorithm's effectiveness.

Weaknesses: It is unclear whether the regret bounds are nearly optimal, as no lower bounds have been established.

**Questions For Authors:**

Is it necessary for the algorithm to know the number of clusters in advance?

**Relation To Broader Scientific Literature:**

Clustering bandits have been extensively studied in the literature, while this paper is the first to explore clustering bandits with dueling feedback.

**Theoretical Claims:**

I have reviewed the proof, which appears to be correct.

---

> ### Author Rebuttal · Authors · 2025-04-01
>
> Thanks for your positive feedback and valuable suggestions, our responses are as follows.
>
> **Q1:** Lower bounds:
>
> **A1:** Thanks for the helpful suggestion. Based on prior techniques (Wang et al., 2024a; Liu et al., 2022) and the single-user dueling bandit lower bound (Saha, 2021), we can derive a lower bound of $O(\sqrt{dmT})$ for the linear setting. Our Algo.1 achieves an upper bound of $O(d\sqrt{mT})$, which is tight up to a $\sqrt d$ factor—a common gap in linear bandits (e.g., LinUCB). Thus, our upper bound is tight and optimal in $m$ for the linear case, and we will include this lower bound in the final version.
>
> For the neural case, even the single-user non-dueling setting lacks a known lower bound, making it an open problem. However, we hypothesize that our upper bound is also optimal in $m$ for the neural setting, as clustering naturally introduces a $\sqrt m$ scaling compared to the single-user case.
>
> **Q2:** Technical challenges involved in the algorithm design and its analysis:
>
> **A2:**
>
> **Novel Algorithm Design:**
> - Cluster Estimation under Dueling Feedback: Our setting does not allow direct use of prior cluster vector estimation methods in classical CB with numerical feedback. With numerical feedback, closed-form ridge regression enables simple updates of user-level covariance matrices and regressors, followed by cluster vector computation using aggregated statistics (e.g., Lines 6–8 in Algo.1 of Wang et al., 2024a). In contrast, dueling feedback lacks closed-form MLE, making direct adaptation infeasible. To address this, we maintain a history buffer $\mathcal{D}_t$ (e.g., Line 10, Algo.1) recording user indices, arm choices, and dueling feedback. We then filter data for users in inferred clusters and estimate cluster vectors via MLE (e.g., Line 6, Algo.1).
>
> - Carefully Designed Edge Deletion Thresholds: Threshold design under dueling feedback is more intricate than in absolute feedback settings and relies on refined calculations (Lemmas B.2 and C.5) to ensure correctness. In the neural setting, this is further complicated by the non-linear reward function. The required Cluster Separation assumption (Assumption 2.7) deviates from those in prior CB works, making its integration into the edge deletion rule for CONDB particularly challenging.
>
> **Novel Theoretical Analysis**:
> We highlight the main technical challenges for CONDB as an example:
> - Incorporating the novel Cluster Separation assumption (Assumption 2.7) for CONDB into the analytical framework of linear CB poses substantial difficulties. We address this by leveraging a linear approximation of the neural net as the analytical basis (Lemmas C.1 and C.2).
> - Deriving $T_0$ after which all users are correctly clustered (Lemma C.5), is particularly challenging due to the **non-linear and dueling nature of the reward**. Classical CB analysis bounds the L2 distance between true and estimated **linear parameters**, which is not applicable here. To overcome this, we use neural network linearization and rigorously analyze the resulting approximation error (Lemmas C.3, C.4, and C.5).
> - A key challenge in proving a sub-linear bound lies in the proper use of the effective dimension $\widetilde d$ (Eq.(16)). Direct extensions of Lemma B.5 from the COLDB (linear rewards) analysis would yield a bound linearly dependent on $p=O(m_{\text{NN}}^2)$, which becomes vacuous as $m_{\text{NN}}$ grows polynomially with $T$. To address this, we adopt techniques from neural bandits to reformulate Lemma B.5 for CONDB, allowing us to express the regret in terms of $\widetilde d$ (Lemma C.9).
>
> **Q3:** Is it necessary for the algorithm to know the number of clusters in advance?
>
> **A3:** No, our algorithms can adpatively cluster the users during the learning process, the number of clusters $m$ does not need to be known.

---

### Official Review · Reviewer_WQTH · 2025-03-15

**Overall Recommendation:** 3

**Summary:**

The paper introduces the first algorithms for clustering users in dueling bandit settings where feedback is based on preferences between pairs of items rather than absolute numerical rewards. The authors propose two novel approaches:

- Clustering of Linear Dueling Bandits (COLDB) for linear reward functions

- Clustering of Neural Dueling Bandits (CONDB) for non-linear reward functions modeled by neural networks.

Both algorithms adaptively group users with same preferences into clusters, allowing collaboration by sharing data within clusters.

The authors provide theoretical analysis showing that both algorithms achieve sub-linear regret bounds that improve when more users belong to the same cluster. Their analysis quantifies the benefits of cross-user collaboration in preference-based feedback scenarios. Experimental results on both synthetic and real-world (MovieLens) datasets demonstrate that the proposed methods outperform independent dueling bandit baselines, validating the theoretical findings.

**Claims And Evidence:**

Most claims are supported by theoretical analysis and empirical evaluation, but with some limitations:

The theoretical regret bounds for both algorithms are well-established with detailed proofs. The authors show that when clustering works correctly, the regret scales with the number of clusters (m) rather than the number of users (u), providing a formal justification for clustering benefits.

However, the empirical evidence is somewhat limited. While the experiments show improved performance over independent baselines, the paper only evaluates two datasets with a small number of simulation settings. The experimental section lacks ablation studies to understand the contribution of different algorithm components, sensitivity analyses for key parameters, and comprehensive evaluations across diverse scenarios.

The claim of correct clustering convergence is supported theoretically (Lemma B.2 for COLDB and Lemma C.5 for CONDB), but these critical lemmas are barely mentioned in the main paper. Additionally, there is confusion around the edge removal mechanism and its notation, as the function 'f' is not clearly defined when discussing step 8 in both algorithms yet plays a central role in the proofs.

Not sure if this is the right place to mention :
- The problem setup of Same preference seems is quite restricted and may limit real-world applicability. A setup with similar preference (same preference with some error tolerance) might be more lucrative.
- The paper repeatedly refers to "cross-user collaboration," "user collaboration," and similar terms, but this terminology is seems slightly misleading. What the paper actually describes is centralized information pooling and model sharing managed by a single algorithm, not active communication between users. Users are passive participants, and all "collaboration" is implicitly facilitated through the central algorithm. The authors should clarify this distinction to avoid misrepresenting the nature of their contribution, especially since true cross-user communication would involve different theoretical and practical challenges.
- The algorithms assume a central entity with access to all user data, uniform user arrivals, discrete and fixed clusters of users, and the ability to run potentially expensive computations for every interaction. Many practical applications, especially those involving privacy concerns or distributed systems, would violate these assumptions. The paper would be strengthened by discussing these limitations and how the approach might be adapted to more realistic constraints.

**Essential References Not Discussed:**

The paper has a comprehensive literature review covering most relevant work. There are few plus minuses which can happen, but that is largely an author's choice.

**Experimental Designs Or Analyses:**

Potential Concerns: While the simulations serve as a great PoC with a limited setup. I would add the following critique:

- The experiments only test two cluster configurations (m=2 and m=5) in the synthetic setting, which doesn't sufficiently validate the algorithms' performance across varying clustering structures.

- The neural network experiments use only a simple square reward function (f(x) = (θᵀx)²). Testing more complex non-linear functions would better demonstrate CONDB's capabilities. At this point, the simulations seem to serve a PoC rather than to verify that the algorithm is behaving as expected.

-  No sim results focussing on the convergence of the clusters.

- There's no analysis of how algorithm parameters (like edge deletion thresholds) affect performance or clustering accuracy.

- The paper doesn't report confidence intervals or statistical significance of the results, making it difficult to assess the reliability of the performance differences.

- There's no evaluation of computational efficiency, which is especially concerning for the neural network-based approach that requires retraining models repeatedly.

- The setup may be unrealistic for practical applications - particularly CONDB's requirement to run neural networks twice per round, which would be computationally prohibitive in many real-world settings.

Any addressing of the above points would be a great help to any reader.

**Methods And Evaluation Criteria:**

The proposed methods -- CONDB and COLDB -- build upon established techniques in bandits literature and make sensible extensions to the preference-feedback setting. The graph-based clustering approach with adaptive edge deletion is reasonable for identifying user groups with ***same*** preferences.

However, there are substantial concerns about practicality. The paper's motivation mentions scenarios with large numbers of options, but CONDB requires running neural networks for each round twice for every round of the for-loop, which is computationally expensive and not scalable for real-world applications. This mismatch between motivation and implementation raises questions about real-world applicability.

The evaluation criteria involving cumulative regret is standard and appropriate for the bandit setting, but the experimental evaluation is limited in several ways:
- Only two cluster configurations (m=2 and m=5) are tested, which doesn't fully validate the algorithms' performance across varying clustering structures.
- The uniform user arrival assumption in the theoretical analysis is unlikely to hold in real applications. Atleast for the simulations, maybe a broader user arrival strategy could be showcased.
- The paper lacks comparison against additional baselines beyond independent dueling bandits.
- No evaluation of computational efficiency is provided, which is particularly concerning for the neural network approach. (minor point)

**Other Comments Or Suggestions:**

**Suggestions/ Comments:**

- Mentioning Lemmas B.2 and C.5 (on edge deletion) to the main text would be very beneficial. These are critical for algorithm correctness but barely mentioned.

- Would it be possible to clarify the notation for the threshold function 'f' used in the edge deletion criteria, as it is inconsistently defined.

- The introduction mentions applications to prompt optimization for large language models. This is very interesting use case but isn't explored further.

- Ablation studies (this would include other non-linear reward structures) to understand the contribution of different algorithm components would be a crucial add.

- Please Provide confidence intervals or statistical significance for experimental results.

**Other Strengths And Weaknesses:**

**Strengths:**

- The paper addresses a practically relevant problem of enabling collaboration among users in preference-based feedback settings.

- The theoretical analysis provides insight into the benefits of clustering in dueling bandit settings.

- The extension to neural networks for handling non-linear reward functions acknowledges the need for more expressive modeling of application setups.

**Weaknesses:**

- The paper appears to deriving a lot of the existing techniques from different areas rather than introducing fundamentally new algorithmic approaches.

- The edge deletion mechanism lacks sufficient analysis on its robustness, particularly in early stages when estimates have high uncertainty.

- The practical implementation of CONDB might face computational challenges that aren't addressed. Training neural networks twice per round is typically expensive for real-world applications.

- The motivation discussing large-scale recommendation systems with many options conflicts with the proposed methodology's computational requirements.

- Missing lower bounds make it difficult to assess whether the derived regret rates are optimal.

- The experimental section lacks depth and breadth needed to fully validate the practical performance of the proposed algorithms.

- The NTK-based theoretical guarantees require extremely wide neural networks. Atleast a simulation which shows it works with smaller nets would go a long way

**Questions For Authors:**

Would like the authors to expand on just one point:

- Your paper frames the approach as enabling "cross-user collaboration," but all information sharing is mediated through a central algorithm with no direct user-to-user communication. How would your approach change if users could only share limited information, had privacy constraints, or were distributed across different systems without centralized coordination?

**Relation To Broader Scientific Literature:**

The paper appears to largely combine existing approaches from different areas rather than introducing fundamentally new techniques. It bridges previously separate research areas:

- The clustering of bandits approach builds on the work of Gentile et al. (2014) -- CLUB algorithm, which introduced graph-based user clustering for standard (non-dueling) contextual bandits.

- The dueling bandits aspect draws from recent work on contextual dueling bandits which established theoretical foundations for linear dueling bandits.

- The CONDB algorithm integrates ideas from neural bandits literature and recent work on neural dueling bandits (Verma et al., 2024).

While the paper cites relevant literature and contributes by combining these existing approaches.

**Theoretical Claims:**

I skimmed over the main theoretical results (Theorems 4.1 and 4.2) and their supporting lemmas. I did give more focus on Lemma B.2, C.5 since they seem central to the paper.

The proofs appear technically sound in deriving the regret bounds, but there are notable limitations:

- The paper provides upper bounds on regret but no lower bounds. Without lower bounds, it's unclear whether the derived regret rates are optimal in terms of dimensionality dependence.

- The edge deletion mechanism's theoretical guarantees (Lemma B.2 and C.5) don't fully address how the algorithms recover from incorrect edge deletions that may occur early in the process when estimates have high uncertainty. In the worst case of incorrect deletions (early in the algorithmic run or out of high probability zone), what is the worst to expect?

- The notation for the function 'f' used in the edge deletion criteria (step 8 in both algorithms) is inconsistently defined between the main text and proofs, creating confusion about this critical component.

- For CONDB, the NTK approximation in Lemma C.1 requires neural networks to be extremely wide (equation 66 specifies polynomial dependencies on multiple parameters), raising concerns about theoretical vs. practical guarantees.

- The regret decomposition in equation (82) assumes correct clustering but doesn't fully account for potential cascading effects from early misclassifications.

---

> ### Author Rebuttal · Authors · 2025-04-01
>
> Thanks for your valuable suggestions. Our responses are as follows. We will incorporate them after revision.
>
> **Q1**: Lemma B.2, C.5, and more robust deletion:
>
> **A1**: We will emphasize these lemmas in the main text. Our edge deletion mechanism and analysis follow prior clustering of bandits (CB) works. For stronger early-stage robustness, the algorithms can be modified to restart with a complete graph each round and re-check edges involving previously served users. This ensures correct clustering w.h.p. after $T_0$, mitigating early-stage misclusterings.
>
> **Q2:** Definition of f:
>
> **A2**: We have defined f at the Input of Algo.1 (top of page 6) it's used in the proof without restating. We will redefinie f in the proof.
>
> **Q3:** Setting of similar preference in clusters:
>
> **A3:** As the first to study preference feedback in CB, we focus on standard assumptions to address its core challenges. While similar-preference settings with numerical feedback have been studied (Ban & He, 2021b; Wang et al., 2024b), extending our results to that setting is an independent direction which we leave for future work.
>
> **Q4:** Terminology of "user collaboration" and the question about the limited shared information setting:
>
> **A4**:
> - We will replace “user collaboration” with clearer terms like “user relation” or “user similarity” as per your advice.
> - For scenarios without centralized coordination or with limited shared information, these fall within the scope of federated bandits. Since our work follows the standard CB setting, such extensions are beyond our current scope, but we view this as a promising future direction.
>
> **Q5**: Assumptions (particularly uniform arrival):
>
> **A5**: Our assumptions follow standard practice in CB and are used solely for theoretical analysis. The uniform arrival assumption can be relaxed to general distributions (Lines 161–164).
>
>
> **Q6:** Computation of CONDB:
>
> **A6:** Even in the baseline where each user runs NDB independently, an NN must still be trained per iteration—yet this incurs higher regret (scaling with $\sqrt u$). Our CONDB trades off computation for better performance. To reduce computational overhead, we can adopt batched training strategies (e.g., "Batched Neural Bandits" (Gu et al., 2024)), updating the NN only every few iterations.
>
> **Q7:** Techinical challenges
>
> **A7:** The dueling (and neural) feedback introduces new technical challenges to the classic CB study. Please kindly refer to our **A1 in response to Reviewer AU83** for details.
>
> **Q8:** Lower bounds:
>
> **A8:** Following Wang et al. (2024a) and Saha (2021), we can derive a lower bound of $O(\sqrt{dmT})$ for linear case. Algo.1 achieves an upper bound of $O(d\sqrt{mT})$, which is tight up to $\sqrt d$—a common gap in linear bandits (e.g., LinUCB). So our upper bound is tight and optimal in $m$ for the linear case.
>
> For the neural case, lower bounds remain open even in the single-user setting. However, we hypothesize that our bound is also optimal in $m$ in the neural setting, as clustering naturally introduces a $\sqrt m$ scaling over the single-user case.
>
> **Q9:** Experiments:
>
> **A9**: Our main contribution is theoretical, but we have added experiments as per your advice to support our findings. Due to time constraints, we leave more extensive experiments as future work.
>
> **(1) Only two datasets**: We added an experiment with the Yelp dataset (see [figure](https://postimg.cc/QHS7rsbP)).
>
> **(2) Ablation for m**: We added an additional value of m=4 (see [figure](https://postimg.cc/qg74j037)), the results are consistent.
>
> **(3) Non-uniform arrival**: We included experiments under non-uniform arrivals, which show consistent performance gains (see [figure](https://postimg.cc/pyr3wNjd)). The user arrival probabilities are randomly sampled from a Dirichlet distribution with a concentration parameter of 10.
>
> **(4) Baselines beyond independent dueling bandits**: Prior CB methods do not handle preference feedback, so direct comparisons are not applicable. Second, following standard CB practice, it is common to use independent per-user algorithms as baselines when introducing a new setting.
>
> **(5) More complex non-linear functions**: We tested a more complex function: $ (\theta^T x)^4 + cos(\theta^T x)$. CONDB still consistently outperforms the baseline (see [figure](https://postimg.cc/1gpyVkBS)).
>
> **(6) Confidence intervals**: We have indeed included confidence intervals (Lines 369-371 and the figures).
>
> **Q10:** NTK:
>
> **A10:** The NTK assumption follows standard practice in neural bandit theory and is used only for analysis; in practice, very wide networks are not needed. Our experiments indeed use small NNs—1 hidden layer with 32 nodes—which already achieve strong performance.
>
> **Q11:** LLM application:
>
> **A11:** The recent work "Prompt Optimization with Human Feedback" uses neural dueling bandits for prompt optimization. Our methods can be directly applied to extend their method to the multi-user setting.

---

> > ### Comment · Reviewer_WQTH · 2025-04-05
> >
> > I would like to thank the authors for their comments. I have increased my score accordingly.

---

> > > ### Author Response · Authors · 2025-04-05
> > >
> > > Thank you for increasing your score! We are pleased to see that our responses addressed your concerns, and we greatly appreciate the constructive feedback you provided. We will incorporate your suggestions to further strengthen and improve our paper.

---

### Decision · Program_Chairs · 2025-05-01

**Decision:**

Accept (poster)

**Comment:**

This paper proposes a new clustering of dueling bandit problem to model the collaborative decision-making scenarios, and proposes two algorithms to solve the problem. The author rebuttals have addressed most of the reviewers' concerns and all reviewers are in favor of accepting this paper. Although we don't have active discussion among reviewers, I recommend acceptance.